# Carrot and Stick:
# Eliciting Comparison Data and Beyond

**Yiling Chen**
Harvard University
yiling@seas.harvard.edu

**Shi Feng**
Harvard University
shifeng@fas.harvard.edu

**Fang-Yi Yu**
George Mason University
fangyiyu@gmu.edu

## Abstract

Comparison data elicited from people are fundamental to many machine learning tasks, including reinforcement learning from human feedback for large language models and estimating ranking models. They are typically subjective and not directly verifiable. How to truthfully elicit such comparison data from rational individuals? We design peer prediction mechanisms for eliciting comparison data using a bonus-penalty payment [11]. Our design leverages on the strong stochastic transitivity for comparison data [60, 13] to create symmetrically strongly truthful mechanisms such that truth-telling 1) forms a strict Bayesian Nash equilibrium, and 2) yields the highest payment among all symmetric equilibria. Each individual only needs to evaluate one pair of items and report her comparison in our mechanism.

We further extend the bonus-penalty payment concept to eliciting networked data, designing a symmetrically strongly truthful mechanism when agents' private signals are sampled according to the Ising models. We provide the necessary and sufficient conditions for our bonus-penalty payment to have truth-telling as a strict Bayesian Nash equilibrium. Experiments on two real-world datasets further support our theoretical discoveries.

## 1 Introduction

In the past two decades, researchers have been embracing the challenge of eliciting private information from individuals when there is no ground truth available to evaluate the quality of elicited contributions, and have made amazing progress. Many mechanisms, collectively called *peer prediction* [42], have been developed to incentivize individuals to strictly truthfully report their information at a Bayesian Nash equilibrium (BNE), by artful design of payment functions that only depend on reports from individuals. Moreover, in multi-task peer prediction mechanisms, the truthful BNE gives each individual the highest expected payoff among all BNEs (i.e. it's a strongly truthful BNE). [11, 57, 30]

However, all prior multi-task peer prediction mechanisms require tasks being ex-ante identical, and hence individuals' private information is independently and identically distributed (iid) for each task. Multi-task peer prediction leverages this structure of information to succeed at truthful elicitation. But what if such structure of information doesn't hold for an information elicitation problem?

One notable application is to elicit pair-wise comparisons of multiple alternatives, such as preferences for consumer products [53], translation [34], peer grading [55], and relevance of language model outputs [9, 10]. Such pair-wise comparison data are crucial for estimating a ranking of the alternatives and for devising reward functions for reinforcement learning. Comparison tasks for different pairs are clearly not ex-ante identical — answers to the tasks demonstrate a certain degree of transitivity (e.g. if $a$ is preferred to $a'$ and $a'$ is preferred to $a''$, then it's more likely that $a$ is preferred to $a''$), rendering existing peer prediction mechanisms not applicable.

38th Conference on Neural Information Processing Systems (NeurIPS 2024).

In this paper, we design a peer prediction mechanism for eliciting comparison data. We model individuals' private information of pair-wise comparisons as Bayesian strongly stochastically transitive (Bayesian SST), which takes many widely used models (e.g. Thurstone [58], Bradley-Terry-Luce [4, 38], and Mallows [39]) as special cases. Our mechanism uses a simple bonus-penalty payment [11] (hence carrot and stick) that takes three reports as inputs and admits a strongly truthful symmetric BNE. The key insight that we develop is a condition of information structure that we call *uniform dominance*. When uniform dominance is satisfied, the bonus-penalty payment is the only type of payment that induces a strictly truthful BNE. Information of individuals, $i$, $j$, and $k$, satisfies uniform dominance if, conditioned on any realization of agent $i$'s information, the probability for $j$'s information to agree with $i$'s is higher than the probability for $k$'s information to agree with $i$'s. Bayesian SST allows us to group three pairwise comparisons, $(a, a')$, $(a'', a')$ and $(a'', a)$, together such that private information about these pairs satisfies uniform dominance. After identifying uniform dominance as a central structure for incentivizing truthful elicitation, we further generalize the bonus-penalty payment to truthfully elicit private information over social networks that demonstrate homophily (i.e. friends tend to have similar opinions than non-friends) [40], and our mechanism can be integrated with common survey techniques such as snowball sampling [24].

**Our contributions.** Our work is a leap forward for designing mechanisms for complex information elicitation settings where ground truth verification is not available.

- We are the first to design mechanisms to truthfully elicit pairwise comparison data under Bayesian SST and networked data under Ising models. In our mechanisms, truthful reporting forms a BNE and yields a strictly higher payoff than any symmetric non-permutation equilibrium.
- We identify a key structure of information, uniform dominance, as a lever such that the simple bonus-penalty payment is the unique payment inducing a strictly truthful BNE. This identification may offer a path for developing truthful elicitation mechanisms for other settings in the future.
- We use Griffiths' inequality and Weitz's self-avoiding walk [65] to prove the uniform dominance property in the Ising model. The resulting correlation bounds may be of independent interest.
- We test our mechanisms on real-world data (sushi preference dataset [26, 27] and Last.fm dataset [8]). Even though these datasets do not perfectly satisfy our theoretical assumptions, our mechanisms still provide a stronger incentive for truthful reporting compared to misreporting.

**Related work.** Information elicitation has two settings according to whether verification is possible. Our paper focuses on elicitation without verification.

For information elicitation without verification, Miller et al. [41] introduce the first mechanism for single task signal elicitation that has truth-telling as a strict Bayesian Nash equilibrium but requires full knowledge of the common prior. Bayesian truth serum (BTS) [47] is the first strongly truthful peer prediction mechanism, but requires complicated reports from agents (their private signal and predictions on other's reports). A series of works [48, 49, 67, 66, 68, 3, 51, 31] relax certain assumptions of BTS but still require complicated reports from agents. Dasgupta and Ghosh [11] introduces the multi-task setting where agents are assigned batch iid tasks and only report their signals. Several works extend this to multiple-choice questions [29, 33, 56, 11], predictions [37], or even continuous value [50], and investigate the limitation and robustness [52, 6, 70, 17, 70]. Another related line of work is co-training and federated learning, which wants to elicit models [32, 36], or samples [64] when multiple iid data or feature of data are available. For more related works, see Faltings [16].

One popular line of work considers information elicitation when verification is possible. Spot-checking requires direct verification of the agent's report [21]. Recent work on comparison data elicitation [19] utilizes spot-checking concepts and focuses on incentivizing effort. Another form of verification involves using additional samples to evaluate how the agent's reports improve model performance [1, 28]. Additionally, the verification may have a general relation to the agent's signal, e.g., proper scoring rules [23, 46, 35, 20].

## 2 Problem Formulation

We discuss our model for eliciting comparison data in this section and defer the extensions to Section 5. Given a collection of items $\mathcal{A}$ and a set of strategic agents $\mathcal{N}$. Agents privately observe

noisy comparisons between pairs of items. Our goal is to design mechanisms to truthfully elicit agents' private information. We will first introduce the information structure of agents' private information of pairwise comparisons in Section 2.1 and then define the information elicitation problem in Section 2.2.

## 2.1 Bayesian SST Models for Comparison Data

We introduce Bayesian Strong Stochastic Transitivity (Bayesian SST) models to capture the structure of agents' private information for comparison data.

Given the set of items $\mathcal{A}$, the underlying unknown state about the items is $\theta \in \Theta$. $\theta$ can be the vector of quality scores for the items (Example 2.2) or a reference ranking (Example 2.4). $\theta$ is drawn according to a common prior $P_\Theta$: $\theta \sim P_\Theta$. Any realized $\theta$ has an associated *stochastic comparison function* $T_\theta : \mathcal{A}^2 \to \{-1, 1\}$. For comparisons of two items $a$ and $a'$, $T_\theta(a, a')$ and $T_\theta(a', a)$ stochastically take value 1 or $-1$, with $\Pr[T_\theta(a, a') = 1] = 1 - \Pr[T_\theta(a', a) = -1]$. For any $\theta$, $T_\theta$ is strongly stochastically transitive as defined below.

**Definition 2.1** ([60, 13]). A stochastic comparison function, $T : \mathcal{A}^2 \to \{-1, 1\}$, is *strongly stochastically transitive (SST)* if for all $a, a', a'' \in \mathcal{A}$ with $\Pr[T(a, a') = 1] > 1/2$ and $\Pr[T(a', a'') = 1] > 1/2$, we have
$$\Pr[T(a, a'') = 1] > \max\{\Pr[T(a, a') = 1], \Pr[T(a', a'') = 1]\}.$$

Intuitively, a comparison function is SST when for any three items $a, a', a''$, if $a$ is more favorable than $a'$ and $a'$ is more favorable than $a''$, then $a$ is even more favorable than $a''$. The concept of SST is a well-established property of comparisons in social science and psychology [18].

Each agent $i \in \mathcal{N}$ has the knowledge of $(T_\theta)_{\theta \in \Theta}$ and $P_\Theta$. When asked to compare a pair of items $(a, a')$, the agent observes an independent draw according to the stochastic comparison function: $S_i = T_\theta(a, a')$, where realization $s_i = 1$ represent item $a$ is preferred over item $a'$ by agent $i$. We assume items are a priori similar but ex-post distinct so that for all $a, a' \in \mathcal{A}$, $\mathbb{E}[T_\theta(a, a')] = \mathbb{E}[\mathbb{E}[T_\theta(a, a') \mid \theta]] = 0$ and $\mathbb{E}[T_\theta(a, a') \mid \theta] \neq 0$ for all $\theta$.

**Examples of Bayesian SST models.** Bayesian SST models are a general family of models that take many classical parametric ranking models, including Bradley-Terry Luce [4, 38], Thurstone (Case V) [58], and Mallows $\eta$-model [39], as special cases.

**Example 2.2** (Bradley-Terry-Luce, Thurstone model, and more [59]). Let $\theta \in \mathbb{R}^{\mathcal{A}} = \Theta$ where each coordinate is independently and identically sampled from a fixed non-atomic distribution $\nu$ on $\mathbb{R}$, and each item $a$ have a scalar quality $\theta_a \in \mathbb{R}$. Let $F : \mathbb{R} \to [0, 1]$ be any strictly increasing function such that $F(t) = 1 - F(-t)$ for all $t \in \mathbb{R}$. Conditional on a fixed $\theta$,
$$\Pr[T_\theta(a, a') = 1] = F(\theta_a - \theta_{a'}) \text{ for all } a, a' \in \mathcal{A}.$$

This model recovers the Thurstone model [58] by setting $F(t) = \Phi(t)$ where $\Phi$ is the Gaussian CDF, and the Bradley-Terry-Luce model [4] by setting $F(t) = \frac{e^t}{1+e^t}$, the sigmoid function. Moreover, this model also contains any additive random utility model [2] where $T(a, a') = 1$ if $\theta_a + Z > \theta_{a'} + Z'$ with iid noise $Z$ and $Z'$, because we can set $F$ to be the CDF of the difference of two iid noise.

**Proposition 2.3.** *For any strictly increasing $F$ and non-atomic $\nu$ on $\mathbb{R}$, the parametric model in Example 2.2 is a Bayesian SST model.*

**Example 2.4** (Mallows $\eta$-model [39]). Let $\Theta$ be the set of rankings on $\mathcal{A}$ and $\eta > 0$ be a dispersion parameter. Given a reference ranking $\theta \in \Theta$, the *Mallows $\eta$-model* generate a ranking $\phi \in \Theta$ with probability $\Pr(\phi) \propto \exp(-\eta d(\theta, \phi))$ where $d(\theta, \phi) = |\{(a, a') \in \mathcal{A}^2 : \theta(a) < \theta(a') \text{ and } \phi(a) > \phi(a')\}|$ is Kendall's tau distance, and $\theta(a)$ is the rank of item $a$. Therefore, to generate comparisons, we first sample a uniform $\theta$ and
$$\Pr[T_\theta(a, a') = 1] = \sum_{\phi : \phi(a) > \phi(a')} \Pr(\phi), \text{ for all } a, a' \in \mathcal{A}.$$

**Proposition 2.5.** *For any $\eta > 0$, Mallows $\eta$-model in Example 2.4 with uniform distribution on reference ranking is an Bayesian SST model.*

The proofs for propositions 2.3 and 2.5 are closely related to strong stochastic transitivity [54, 7], but are provided in the appendix for completeness.

## 2.2 Peer Prediction Mechanism Design

To truthfully elicit comparison data from agents, a peer prediction mechanism creates a game between the agents outlined below: First, we choose an *assignment* $\mathcal{E} = \{e_i = (a_{u_i}, a_{v_i}) : i \in \mathcal{N}\}$ where agent $i \in \mathcal{N}$ gets a pair of items $e_i = (a_{u_i}, a_{v_i}) \in \mathcal{A}^2$ to compare. Then each agent $i \in \mathcal{N}$ privately observes the realization of the comparison (signal) $s_i \in \{-1, 1\}$, which is an independent realization of $T_\theta(a_{u_i}, a_{v_i})$, and reports $\hat{s}_i \in \{-1, 1\}$ potentially different from her signal. We use $S_i = S(a_{u_i}, a_{v_i})$ to denote the random variable of agent $i$'s signal, where the randomness of $S(\cdot, \cdot)$ comes from both $\theta$ and $T_\theta$. Let $\mathbf{S}$ represent the random vector of all agents' signals, $\mathbf{s} = (s_i)_{i \in \mathcal{N}}$ be all agents' realized private signals and $\hat{\mathbf{s}} = (\hat{s}_i)_{i \in \mathcal{N}}$ be all agents' reports. Finally, a *peer prediction mechanism* $(M_i)_{i \in \mathcal{N}}$ takes all agents' reports $\hat{\mathbf{s}}$ and pays agent $i$ with $M_i(\hat{\mathbf{s}}) \in \mathbb{R}$.

Each agent $i$'s strategy is a random function from her signal to a report $\sigma_i : s_i \mapsto \hat{s}_i$, and the randomness of their strategies is independent of each other's and all signals. With slight abuse of notation, we write $\sigma_i(s_i, \hat{s}_i) = \Pr[\hat{S}_i = \hat{s}_i \mid S_i = s_i]$ as the conditional probability of reporting $\hat{s}_i$ given private signal $s_i$. A strategy profile $\sigma$ is a collection of all agent's strategies. All agents are rational and risk-neutral, so they want to maximize their expected payments. Thus, given prior $P_\Theta$, randomness of $T_\theta$ and a strategy profile $\sigma$, agent $i$ wants to maximize her ex-ante payment denoted as $\mathbb{E}_{\sigma, \theta, T_\theta}[M_i(\hat{\mathbf{S}})]$ where $\hat{\mathbf{S}}$ is the random vector of all agents' report that depends on the signals $\mathbf{S}$ and strategy profile $\sigma$.

We introduce three families of strategies, truth-telling, permutation, and uninformed strategy profiles, which are central to understanding effective peer prediction mechanisms.

- A strategy $\sigma_i$ is *truthful* (or truth-telling) if it is a deterministic identity map, $\sigma_i(s_i) = s_i$. A strategy profile is truthful if all agents' strategies are truthful.
- A *permutation strategy profile* is where agents simultaneously relabel their signals and then report the relabeled ones. A permutation strategy is indistinguishable from truth-telling unless the peer prediction mechanism has additional knowledge about the prior signal distribution.[33]
- Finally, a strategy is *uninformed* if it has the same report distribution across all signals, and it is informed otherwise. Common examples include consistently reporting all signals as a constant value, such as 1 or $-1$, or using a random report regardless of the signal. Uninformed strategies are undesirable as the reports bear no relationship to the private signals.

A strategy profile is *symmetric* if all agents use the same strategy. For example, both truth-telling and permutation strategy profiles are symmetric.

We now introduce goals for a peer prediction mechanism that favors truth-telling more than other strategies. First, we want the truth-telling (strategy profile) to be a strict *Bayesian Nash equilibrium (BNE)* so that any agent's payment would strictly decrease if she unilaterally changes to any non-truthful strategy. Moreover, there may be multiple equilibria, and a desirable mechanism should ensure that truth-telling is better than all other equilibria. In this paper, we aim for symmetrically strongly truthful mechanisms defined below.

**Definition 2.6.** A peer prediction mechanism is **symmetrically strongly truthful** if truth-telling is a BNE, and each agent's expected payment in truth-telling is no less than the payment in any other symmetric equilibrium with equality for the equilibrium with a permutation strategy profile.[1]

## 3 Bonus-penalty Payment Mechanism for Comparison Data

We now propose a **bonus-penalty payment mechanism** for eliciting comparison data. The mechanism makes use of a *bonus-penalty payment* function, which can be seen as an agreement payment and introduced by Dasgupta and Ghosh [11] in a different context (see discussion in appendix A). Formally, for any $\hat{s}_i, \hat{s}_j, \hat{s}_k \in \{-1, 1\}$, the bonus-penalty payment function is

$$U^{BPP}(\hat{s}_i, \hat{s}_j, \hat{s}_k) = \hat{s}_i\hat{s}_j - \hat{s}_i\hat{s}_k = 2\left(\mathbf{1}[\hat{s}_i = \hat{s}_j] - \mathbf{1}[\hat{s}_i = \hat{s}_k]\right), \tag{1}$$

which rewards when the first input agrees with the second but punishes when it agrees with the third.

---

[1]Kong and Schoenebeck [30] shows that it is impossible to pay the truth-telling strategy profile strictly better than other permutation strategy profiles without additional knowledge of the prior signal distribution.

Mechanism 1 uses the bonus-penalty payment eq. (1) for each agent $i$ by carefully choosing agent $j$ and $k$ such that agent $j$'s signal is more likely to agree with agent $i$'s than agent $k$'s signal is. The crux of finding such pair of agents is to show that if agent $i$ prefers item $a$ over $a'$, she would expect that others will prefer any third item $a''$ over $a'$, and prefer $a$ over $a''$. Thus, if agent $j$ has pair $(a'', a')$ and agent $k$ has pair $(a'', a)$, then agent $j$'s signal is more likely to take the same value as $i$'s than agent $k$'s signal is. This is the main idea behind the proof of Theorem 3.1, where we establish the symmetrically strongly truthfulness of Mechanism 1. To ensure the existence of such pairs are assigned, we require the assignment $\mathcal{E}$ to be **admissible** where for all $(a, a') \in \mathcal{E}$, there exists $a'' \in \mathcal{A}$ so that $(a'', a')$ and $(a'', a) \in \mathcal{E}$.

---

**Mechanism 1:** BPP mechanism for comparison data

---

**Input:** Let $\mathcal{A}$ be a collection of items, $\mathcal{E}$ be an admissible assignment, and $\hat{s}$ be agents' reports.

**for** *agent $i \in \mathcal{N}$ with pair $e_i = (a_{u_i}, a_{v_i}) = (a, a')$* **do**

Find $a'' \in \mathcal{A}$ and two agents $j$ and $k$ so that $e_j = (a'', a')$ and $e_k = (a'', a)$, and pay agent $i$

$$M_i(\hat{\mathbf{s}}) = U^{BPP}(\hat{s}_i, \hat{s}_j, \hat{s}_k) = \hat{s}_i \hat{s}_j - \hat{s}_i \hat{s}_k. \tag{2}$$

---

**Theorem 3.1.** *Given a collection of items $\mathcal{A}$ and a set of agents $\mathcal{N}$ with $|\mathcal{A}|, |\mathcal{N}| \geq 3$, for any admissible assignment matrix $\mathcal{E}$ and Bayesian SST model with $(T_\theta)_{\theta \in \Theta}$ and $P_\Theta$, the BPP mechanism for comparison (Mechanism 1) is symmetrically strongly truthful.*

We defer the proof of theorem 3.1 to section 4. The admissible condition imposes little overhead on downstream learning problems, including rank recovery [25] and identification of the top $k$ items [15]. Specifically, the size of assignment $\mathcal{E}$ is the number of comparisons and corresponds to the sample complexity for these learning problems. If a learning algorithm requires a set of pairs to compare $\mathcal{E}^{ML}$, we can construct an admissible superset $\mathcal{E}$ that introduces a constant factor overhead and can recover $\mathcal{E}^{ML} \subseteq \mathcal{E}$.[2]

We remark that the bonus-penalty payment function eq. (2) can be seen as a boolean function for *transitivity* [45]; see remark 3.2 for a formal statement. Hence, theorem 3.1 implies that agents' manipulations can only decrease the probability of transitivity among their reports.

**Remark 3.2.** Note that a deterministic comparison function $t : \mathcal{A} \times \mathcal{A} \to \{-1, 1\}$ satisfies transitivity on three items $a, a', a'' \in \mathcal{A}$ if and only if $t(a, a'), t(a', a''), t(a'', a)$ are not all equal, that is $NAE(t(a, a'), t(a', a''), t(a'', a)) = 1$ where

$$NAE(w_1, w_2, w_3) = \frac{3}{4} - \frac{1}{4} w_1 w_2 - \frac{1}{4} w_1 w_3 - \frac{1}{4} w_2 w_3.$$

The agent's random noisy comparisons may or may not satisfy transitivity. The probability of transitivity is the probability that they do.

We can show that the bonus-penalty payment in eq. (2) is equivalent to the above transitivity test when agents are truth-telling. Formally,

$$NAE(S(a, a'), S(a', a''), S(a'', a))$$
$$= \frac{3}{4} - \frac{1}{4} \left( S(a, a') S(a', a'') + S(a, a') S(a'', a) + S(a', a'') S(a'', a) \right)$$
$$= \frac{1}{4} \left( S(a, a') S(a'', a') - S(a, a') S(a'', a) \right) + \frac{3}{4} + \frac{1}{4} S(a'', a') S(a'', a) \quad (S(a', a'') = -S(a'', a'))$$
$$= \frac{1}{4} \left( s_i s_j - s_i s_k \right) + \frac{3}{4} + \frac{1}{4} s_j s_k = \frac{1}{4} M_i(\mathbf{s}) + \frac{3}{4} + \frac{1}{4} s_j s_k \quad \text{(truth-telling)}$$

Therefore, $\arg\max_{\hat{s}_i} \mathbb{E}[NAE(\hat{s}_i, -S_j, S_k)|S_i = s_i] = \arg\max_{\hat{s}_i} \mathbb{E}\left[ \frac{1}{4} M_i(\hat{s}_i, S_j, S_k) + \frac{3}{4} + \frac{1}{4} S_j S_k | S_i = s_i \right] = \arg\max_{\hat{s}_i} \mathbb{E}[M_i(\hat{s}_i, S_j, S_k)|S_i = s_i]$.

---

[2]Specifically, given any assignment $\mathcal{E}^0$, we can construct a superset $\mathcal{E}$ so that for any $(a, a') \in \mathcal{E}^0$ find an arbitrary $a'' \neq a, a'$ and add $(a, a'), (a', a), (a, a''), (a'', a), (a', a''), (a'', a')$ into $\mathcal{E}$. Thus, $\mathcal{E}$ is admissible and at most six times larger than $\mathcal{E}^0$.

# 4 Proof of Theorem 3.1: from Bayesian SST Model to Uniform Dominance

To prove theorem 3.1, we formalize the idea that agent $j$'s signal is more likely to agree with agent $i$'s than agent $k$'s is as what we call uniform dominance in definition 4.1. We'll show that any Bayesian SST model satisfies this property. Then, we'll prove that BPP mechanism is symmetrically strongly truthful when agents' private signals satisfy uniform dominance.

**Definition 4.1.** Given a random vector $(S_i, S_j, S_k) \in \{-1, 1\}^3$ with joint distribution $P$, $S_j$ **uniformly dominates** $S_k$ for $S_i$ if $\Pr[S_j = 1 \mid S_i = 1] > \Pr[S_k = 1 \mid S_i = 1]$ and $\Pr[S_j = -1 \mid S_i = -1] > \Pr[S_k = -1 \mid S_i = -1]$. We call such an ordered tuple $\langle S_i, S_j, S_k \rangle$ a uniformly dominant tuple.[3]

Lemma 4.2 shows how to identify uniformly dominant tuples under Bayesian SST models.

**Lemma 4.2.** *Under any Bayesian SST model, for any agent i and items a, a′ and a″, agent j's signal $S_j = S(a'', a')$ uniformly dominates agent k's signal $S_k = S(a'', a)$ for signal $S_i = S(a, a')$.*

In other words, under any Bayesian SST model, the distribution of $S(a, a'), S(a'', a'), S(a'', a)$ satisfies uniform dominance for any $a, a', a''$. In the rest of this section, we can view $(S_i, S_j, S_k)$ as an abstract random vector with some joint distribution $P$.

We now establish some implications of uniform dominance on the bonus-penalty payment. Lemma 4.3 shows that truth-telling is the best response if other signals are reported truthfully. Lemma 4.4 states that the expected payment is zero if everyone uses uninformed strategies (random functions independent of input). Lemma 4.5 characterizes the best response under symmetric strategy profiles (the same random function on each coordinate).

**Lemma 4.3** (Truthfulness). *Given a uniformly dominant tuple $\langle S_i, S_j, S_k \rangle$ with distribution P, for all $s_i \in \{-1, 1\}$, $s_i = \arg\max_{\hat{s}_i \in \{-1,1\}} \mathbb{E}_P \left[ U^{BPP}(\hat{s}_i, S_j, S_k) \mid S_i = s_i \right]$ and $\mathbb{E}_P \left[ U^{BPP}(S_i, S_j, S_k) \right] > 0$.*

**Lemma 4.4.** *Given a uniformly dominant tuple $\langle S_i, S_j, S_k \rangle$, with joint distribution P if agent j and k both use an uninformed strategy $\sigma$ so that $\hat{S}_j = \sigma(S_j)$ and $\hat{S}_k = \sigma(S_k)$, for all $s_i$ and $\hat{s}_i$ in $\{-1, 1\}$, $\mathbb{E}_{\sigma,P} \left[ U^{BPP}(\hat{s}_i, \hat{S}_j, \hat{S}_k) \mid S_i = s_i \right] = 0$.*

**Lemma 4.5.** *Given a uniformly dominant tuple $\langle S_i, S_j, S_k \rangle$ with distribution P, for any strategy $\sigma$ and $s_i \in \{-1, 1\}$ when agent j and k both use $\sigma$, $\arg\max_{\hat{s}_i \in \{-1,1\}} \mathbb{E}_{\sigma,P} \left[ U^{BPP}(\hat{s}_i, \hat{S}_j, \hat{S}_k) \mid S_i = s_i \right] = \arg\max_{\hat{s}_i \in \{-1,1\}} \{\sigma(s_i, \hat{s}_i) - \sigma(-s_i, \hat{s}_i)\}$.*

We'd like to highlight that lemmas 4.3 to 4.5 as well as the proof of theorem 3.1 below hold for any uniformly dominant tuple $\langle S_i, S_j, S_k \rangle$, not necessarily derived from the Bayesian SST model. This offers a path to generalize our mechanism for comparison data to other settings.

*Proof of theorem 3.1.* By lemma 4.2, for any agent $i$, the associated agent $j$'s signal $S_j = S(a'', a')$ uniformly dominates the associated $k$'s signal $S_k = S(a'', a)$ for signal $S_i = S(a, a')$. By lemma 4.3, if agent $j$ and $k$ are truthful, agent $i$'s best response is truthful reporting, so truth-telling is a BNE.

Now we show that all other symmetric equilibria are permutation or uninformed equilibria. For any symmetric equilibrium $\sigma = (\sigma_\iota)_{\iota \in \mathcal{N}}$ so that everyone uses the same strategy $\sigma_\iota = \sigma$ for all $\iota \in \mathcal{N}$. If $\sigma$ is not deterministic so that $\sigma(s, s), \sigma(s, -s) > 0$ for some $s \in \{-1, 1\}$, agent $i$ must be indifferent between reporting $s$ and $-s$ when getting $S_i = s$. $\sigma(s, s) - \sigma(-s, s) = \sigma(s, -s) - \sigma(-s, -s)$ by lemma 4.5. This means $\sigma(s, s) = \sigma(-s, s)$ and $\sigma(s, -s) = \sigma(-s, -s)$, and $\sigma$ is an uninformed strategy. If the strategy is deterministic, there are two cases. If $\sigma(s) = \sigma(-s)$, the strategy is also uninformed. If $\sigma(s) \neq \sigma(-s)$, $\sigma$ is either truth-telling $s \mapsto s$ or flipping $s \mapsto -s$ for all $s$.

Finally, by lemma 4.4, any uninformed equilibrium's expectation is zero. Additionally, because eq. (1) is invariant when all inputs are flipped, the truth-telling and flipping/permutation equilibria has the same expected payment which is positive by lemma 4.3. □

---

[3]If we view $S_j$ and $S_k$ as two statistical tests for a binary event $S_i$, the two inequalities in definition 4.1 say that $S_j$ has a better type II and type I error than $S_k$ respectively.

# 5 Generalization of Bonus-penalty Payment Mechanisms

We now leverage the key idea of uniform dominance to design peer prediction mechanisms for networked data in section 5.1. In section 5.2, we summarize our design approach as a general scheme that first identifies uniform dominance structures and then engages the bonus-penalty payment. We prove the uniqueness of bonus-penalty payment: it is the only payment function, up to some positive affine transformation, that induces truth-telling as a strict BNE for all uniform dominant tuples.

## 5.1 Bonus-penalty Payment Mechanisms for Networked Data

Uniform dominance implies agent $i$'s signal is more likely to agree with agent $j$'s than with agent $k$'s. Social networks are another natural domain exhibiting this property, as homophily [40] suggests that agents' opinions or signals in a social network are more likely to agree with their friends than with non-friends. Leveraging this insight, we use a bonus-penalty payment scheme to elicit binary networked data.

---

**Mechanism 2:** BPP mechanism for networked data

**Input:** Let $(V, E)$ be a graph of agents in $V$, $\hat{\mathbf{s}} \in \{-1, 1\}^V$ from all agent's reports.

**for** *agent* $i \in V$ **do**

 Find agents $j$ (friend) and $k$ (non-friend) so that $(i, j) \in E$ but $(i, k) \notin E$, and pay agent $i$

$$M_i(\hat{\mathbf{s}}) = U^{BPP}(\hat{s}_i, \hat{s}_j, \hat{s}_k) = \hat{s}_i \hat{s}_j - \hat{s}_i \hat{s}_k. \tag{3}$$

---

Below, we provide a theoretical guarantee for our mechanism under a popular graphical model for social network data, *Ising model* [14, 43], which captures the correlation between agents and their friends. Formally, an Ising model consists of an undirected graph $(V, E)$ and correlation parameter $\beta_{i,j} \geq 0$ for each edge $(i, j) \in E$. Each agent is a node in the graph, $\mathcal{N} = V$, and has a binary private signal (1 or $-1$) jointly distributed as the following: For all $\mathbf{s} = (s_i)_{i \in V} \in \{-1, 1\}^V$, $\Pr_\beta[\mathbf{S} = \mathbf{s}] \propto \exp(H(\mathbf{s}))$ where the energy function is $H(\mathbf{s}) = \sum_{(i,j) \in E} \beta_{i,j} s_i s_j$.

**Theorem 5.1.** *If agents' signals are sampled from an Ising model on undirected graph $(V, E)$ with correlation parameters $\beta$, Mechanism 2 is symmetrically strongly truthful, when $\frac{2\underline{\beta}}{d} > \ln \frac{e^{2(d+1)\overline{\beta}}+1}{e^{2\overline{\beta}}+e^{2d\overline{\beta}}}$ where $\underline{\beta} = \min_{(i,j) \in E} \beta_{i,j}$, $\overline{\beta} = \max_{(i,j) \in E} \beta_{i,j}$, and $d$ is the maximal degree of graph $(V, E)$.*

Mechanism 2 does not require knowledge about parameters of the Ising model, but only the connection of the network $(V, E)$. Social network platforms, which already possess this knowledge, can easily integrate our mechanism when conducting surveys. Additionally, snowball sampling [24], which relies on participants referring their friends, is also naturally compatible with our mechanism.

The complete proof of theorem 5.1 is quite technical and is deferred to the appendix, where we also explain why the bound between $\beta$ and $d$ is necessary. Below, we provide a sketch of the proof.

*Proof sketch for theorem 5.1.* As discussed in section 4, we only need to show that for any agent $i$, for all agent $j$ with $(i, j) \in E$ and $k$ with $(i, k) \notin E$, $j$'s signal uniformly dominates $k$'s signal for $i$'s signal. Because the energy function $H(\mathbf{s})$ above remains invariant when the signs are flipped, $\Pr[S_i = 1] = \Pr[S_j = 1] = \Pr[S_k = 1] = 1/2$, it is sufficient to prove that

$$\Pr[S_i = 1 \mid S_j = 1] > \Pr[S_i = 1 \mid S_k = 1]. \tag{4}$$

We then prove a lower bound for the left-hand side and an upper bound for the right-hand side separately. For the left-hand side, we use the Griffiths' inequality [44] to show that the minimum value of $\Pr[S_i = 1 \mid S_j = 1]$ happens when $j$ is the only friend of $i$. For the right-hand side, we use Weitz's self-avoiding walk [65] and reduce any graph with maximum degree $d$ into a $d$-ary tree. □

## 5.2 General Design Scheme and Uniqueness

The design of BPP mechanisms for comparison data and networked data has suggested a general design scheme for other elicitation settings. That is, if one can *identify a uniformly dominant tuple for*

**Mechanism 3:** General design scheme using BPP

---

**Input:** Let $\hat{s}$ be reports from agents in $\mathcal{N}$.

**for** *agent* $i \in \mathcal{N}$ **do**

> Find two agents $j$ and $k$ so that $j$'s signal uniformly dominates $k$'s for $i$'s, and pay agent $i$
>
> $$M_i(\hat{s}) = U^{BPP}(\hat{s}_i, \hat{s}_j, \hat{s}_k) = \hat{s}_i\hat{s}_j - \hat{s}_i\hat{s}_k.$$

---

*each agent*, adopting the bonus-penalty payment gives a symmetrically strongly truthful mechanism. We further show that the bonus-penalty payment is in some sense unique.

**Theorem 5.2.** *If for each agent $i$ the associated agent $j$'s signal uniformly dominates $k$'s signal for $i$'s signal, the above scheme is symmetrically strongly truthful.*

When an agent $i$ has multiple pairs of $(j_1, k_1), \ldots, (j_\ell, k_\ell)$ so that $j_l$'s signal uniformly dominates $k_l$'s for $i$'s for each $l = 1, \ldots, \ell$, we may pay agent $i$ the average of bonus-penalty payment on all pairs $M_i(\hat{s}) = \frac{1}{\ell}\sum_{l=1}^{\ell} U^{BPP}(\hat{s}_i, \hat{s}_{j_l}, \hat{s}_{k_l})$. This average maintains our symmetrically strongly truthful guarantee while potentially reducing the variance in payments.

Theorem 5.2 shows that bonus-penalty payment is a sufficient condition for designing good elicitation mechanisms for information structures with uniform dominance. We now prove it is also a necessary condition: any payment that induces truth-telling as a strict BNE under all uniformly dominant tuples must be an affine transformation of the bonus-penalty payment.

**Theorem 5.3** (Uniqueness). *A payment $U : \{-1, 1\}^3 \to \mathbb{R}$ satisfies that, for all uniformly dominant tuples $\langle S_i, S_j, S_k \rangle$, $s_i = \arg\max_{\hat{s}_i \in \{-1,1\}} \mathbb{E}\left[U(\hat{s}_i, S_j, S_k) \mid S_i = s_i\right]$, if and only if there exist $\lambda > 0$ and $\mu : \{-1, 1\}^2 \to \mathbb{R}$ so that*

$$U(s_i, s_j, s_k) = \lambda U^{BPP}(s_i, s_j, s_k) + \mu(s_j, s_k), \text{ for all } s_i, s_j, s_k \in \{-1, 1\}$$

*where choice of $\mu$ does not affect the set of equilibria.*

# 6 Experiments

We present experiments on real-world data to evaluate our models and mechanisms. We hope to cast insights on two questions empirically. Does our mechanism provide better rewards when all agents report truthfully than when all agents report randomly? Does our mechanism incentivize truth-telling for each agent if all other agents are truthful? We evaluate Mechanism 1 and 2 by comparing three settings, truth-telling, uninformed, and unilateral deviation, using *empirical cumulative distribution functions* (ECDF) on agents' payments. Each point on ECDF denotes the fraction of agents who get paid less than a particular value.

For both comparison and networked datasets (figs. 1 and 2), we find our mechanisms provide better payments to agents under truthful settings than the other two settings. The ECDF under truth-telling lies below the other two ECDFs, which is known as first-order stochastic dominance. This implies that the truth-telling strategy results in higher average, quantiles (e.g., first quartile, median, and third quartile), and a greater expectation of any monotone function on the empirical distribution than the other two settings. We provide additional

## 6.1 SUSHI Preference Dataset for Comparison Data

We consider preference data for a collection of 10 sushi items (item set A) [26, 27], and focus on a group of 249 agents. Each agent provides a complete ranking of all 10 types of sushi in the dataset. These agents are female, aged thirty to forty-nine, who took more than three hundred seconds to rank the items and mostly lived in Kanto and Shizuoka until age fifteen. We restrict the set of agents to avoid significant violations of transitivity across different agents and to better align with our model assumptions. In the appendix, we will present the experimental results for other groups of users and further test whether the dataset satisfies transitivity.

For the first question, we use Mechanism 1 to compute each agent's payment under the truth-telling or uninformed strategy profile. For each agent $i$, we 1) randomly sample three items $a, a', a''$ and

two agents $j, k$, 2) derive agent $i$'s comparison on the first two items $(a, a')$ from her ranking, (and similarly for agent $j$'s comparison on $(a', a'')$, and agent $k$'s comparison on $(a, a'')$), 3) compute bonus-penalty payment on these three comparisons, 4) repeat the above procedure 100 times and pay agent $i$ with the average of those 100 trials. For the uninformed strategy setting, we replace every agent's comparisons with uniform random bits and compute the payment. The left column of fig. 1 presents the ECDF of payments for the agents in both settings. The figure shows that in the uninformed random strategy setting only about 50% of the agents receive positive payments, while in the original dataset (truthful strategy setting) over 75% of the users receive positive payments. The right column of fig. 1 tests the second question if the agent has the incentive to deviate when every other agent is truthful. The truth-telling curve is identical to the left column of fig. 1. For unilateral deviation, each agent gets the above bonus-penalty payment when her comparisons are replaced by uniform random bits. We plot the ECDFs of payments for both settings in the right column of fig. 1. The figure shows that the ECDF of the unilateral deviation payments is above the ECDF of human users' payments, indicating that our mechanism pays more to the truth-telling agents.

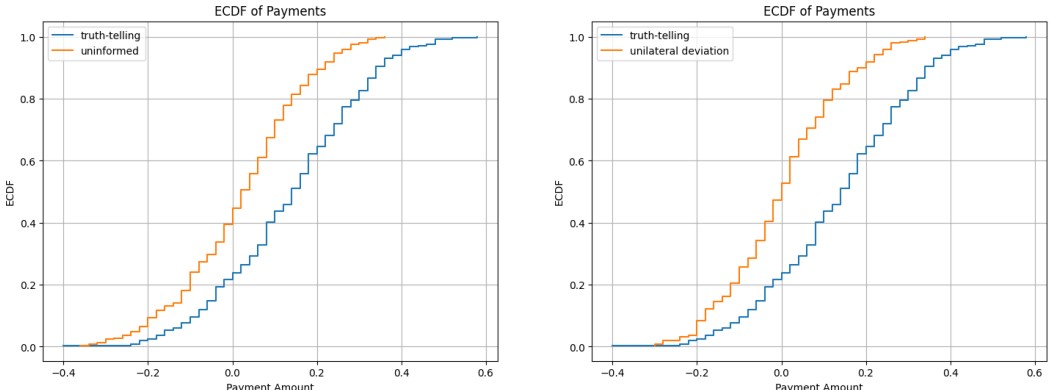

Figure 1: SUSHI preference dataset

## 6.2  Last.fm Dataset for Networked Data

We test our BPP mechanism on the Last.fm dataset from Cantador et al. [8]. This dataset consists of 1892 agents on Last.fm, forming a social network with 12704 edges and an average degree of 6.71. It records agents' top fifty favorite artists whom they have listened to the most. We note that, in the dataset, listener fractions for all artists are much smaller than non-listener fractions. This bias differs from our Ising model in section 5.1 where every agent has the same chance to get both signals. Thus, the result can be seen as a stress test for our mechanism even when the data deviate from the assumption of our theoretical results.

Figure 2 focuses on the most popular artist in the dataset, Lady Gaga, who has a listener fraction of 32.3%. The results for additional artists are presented in the supplementary material. The left column of fig. 2 tests the first question. Each agent has a binary signal about whether or not she listens to a particular artist (Lady Gaga in this section). For the truth-telling setting, everyone reports her signal truthfully and gets payment by the bonus-penalty payment (formally defined in section 5.1). For the uninformed setting, everyone gets the bonus-penalty payment when all reports are iid according to the prior (0.322 for Lady Gaga). When everyone is truthful, more than 76% of agents get positive payments and have an average payment of 0.37 for Lady Gaga, while when agents report randomly, only half get positive payments, and have a near zero average payment. These results suggest that agents got more incentive to choose the truth-telling equilibrium than the uninformed equilibrium. The right column of fig. 2 tests the second question. The truth-telling curve is identical to the left column of fig. 2. For the unilateral deviation setting, each agent gets the bonus-penalty payment when she reports listener/non-listener uniformly at random. The unilateral deviation's payment is worse than the payments for truth-telling, decreasing from 0.37 to near zero for Lady Gaga.

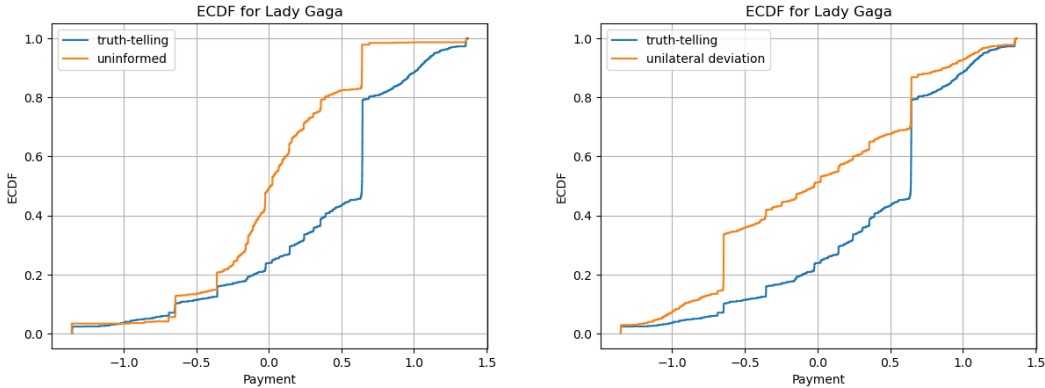

Figure 2: Last.fm dataset for Lady Gaga

# 7 Conclusion and Discussion

We introduce a symmetrically strongly truthful peer prediction mechanism for eliciting comparison data without verification and extend it to eliciting networked data under Ising models. Our mechanisms are evaluated using real-world data. A key insight from our work is the identification of a structure we term "uniform dominance," which suggests a path for designing mechanisms in more complex elicitation settings. For example, in time-series data, adjacent points tend to be more related than distant ones, and in contextual settings, feedback from similar contexts is typically more related than from different contexts.

A central assumption in this study is that agents are *a priori similar*. Hence, noisy comparisons of item pairs are independent of the assigned agent's identity. This assumption is reasonable for items with widely agreed-upon rankings, such as quality assessments of large language model (LLM) outputs. However, it may break down in settings where preferences are highly polarized, such as political opinions or social choice problems[4]. Despite this, our additional experiments in appendix F, which relax the selection rule used in obtaining fig. 1, show that the mechanism remains robust even when some dissimilarities among agents exist.

Agents in our model are assumed to focus solely on maximizing their payments, without accounting for efforts or external incentives such as minimizing others' rewards or intentionally distorting rankings. While our mechanism may be extended to handle binary effort as suggested in previous work [11, 57], accommodating more than two effort levels would require additional assumptions [69]. Moreover, one may hope to incorporate the designer's utility, by factoring in downstream learning problems along with elicitation payments. This would necessitate a significant overhaul of the existing learning framework.

Our mechanisms achieve a symmetric, strongly truthful equilibrium. This does not rule out the existence of non-symmetric equilibria with potentially higher utility. However, such equilibria would require complex coordination among agents, making them less likely to arise naturally.

From a technical standpoint, our approach involves several assumptions that can be generalized or relaxed. Our Bayesian SST model, which relies on strong stochastic transitivity, serves as a non-parametric extension of several widely used parametric ranking models. In appendix C.2, we present both positive and negative results regarding weaker notions of transitivity (e.g., [5]). While we assume admissible assignments, this can be relaxed to random assignments with full support. Additionally, limited liability can be ensured in our mechanism. For example, adding a constant of 1 to the payment function in eq. (2) ensures that the payment is either 2 or 0.

---

[4]For example, when ranking phone features (e.g., innovation, performance, brand reputation, price, ease of use), consumers often fall into two groups: early adopters, who prioritize cutting-edge technology, and late adopters, who favor stability, affordability, and ease of use. Their opposing preferences violate the a priori similarity assumption. Imagine an early adopter whose payment in eq. (2) depends on two late adopters. Since their preferences may differ significantly, the early adopter might have an incentive to misreport her preferences.

## Acknowledgments

This research was partially supported by the National Science Foundation under grant no. IIS-2147187.

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

# A  Further discussion on BPP payment

In this section, we discuss the connection of bonus-penalty payment and existing peer prediction mechanisms. First, if we substitute the third input with a uniformly random bit, denoted as $\hat{s}_k = Z \sim_u \{-1, 1\}$, the bonus-penalty payment simplifies to the *agreement mechanism* [62, 61, 63], one of the most basic peer prediction mechanisms,

$$\mathbb{E}\left[U^{BPP}(\hat{s}_i, \hat{s}_j, Z)\right] = \hat{s}_i \hat{s}_j = 2\mathbf{1}[\hat{s}_i = \hat{s}_j] - 1.$$

However, the agreement mechanism is not symmetrically strongly truthful, as all agents always reporting 1 and $-1$ can result in higher payments than truth-telling.

The bonus-penalty payment eq. (1) is originally proposed by [11, 57] for the multi-task setting. Our BPP mechanism in Mechanism 3 can be seen as a generalization of multi-task setting. In the multi-task setting, agents works on multiple tasks and for each task the private signals are jointly identically and independently (iid) sampled from a fixed distribution and the each agent's strategy also are iid. Take two agents (Isabel and Julia) and two tasks as an example: Isabel has a private signal $(s_i^1, s_i^2)$ and reports $(\hat{s}_i^1, \hat{s}_i^2)$ and Julia has $(s_j^1, s_j^2)$ and reports $(\hat{s}_j^1, \hat{s}_j^2)$ where $(s_i^l, s_j^l)$ are iid from random vector $(S_i, S_j)$. Isabel and Julia decide their reports on each task using random function $\sigma_i, \sigma_j : \{-1, 1\} \mapsto \{-1, 1\}$ respectively. Dasgupta and Ghosh [11] use the following payments for Isabel

$$\mathbf{1}[\hat{s}_i^1 = \hat{s}_j^1] - \mathbf{1}[\hat{s}_i^1 = \hat{s}_j^2] = \frac{1}{2} U^{BPP}\left(\hat{s}_i^1, \hat{s}_j^1, \hat{s}_j^2\right).$$

The payment is a special case of Mechanism 3 by taking the second input as $\hat{s}_j^1$ and the third input as $\hat{s}_j^2$. Additionally, $S_j^1$ uniform dominates $S_j^2$ for $S_i^1$ if and only if

$$\Pr[S_j = 1 \mid S_i = 1] > \Pr[S_j = 1], \text{ and } \Pr[S_j = -1 \mid S_i = -1] > \Pr[S_j = -1]$$

which is called *categorical signal distributions* [57].

Finally, similar to Shnayder et al. [57], we may extend to non-binary signal setting by extending the payment to

$$U^{BPP}(\hat{s}_i, \hat{s}_j, \hat{s}_k) = 2\left(\mathbf{1}[\hat{s}_i = \hat{s}_j] - \mathbf{1}[\hat{s}_i = \hat{s}_k]\right)$$

and the definition of uniform dominance to the following.

**Definition A.1.** Given a random vector $(S_i, S_j, S_k) \in \Omega^3$ on a discrete domain, we say $S_j$ *uniformly dominates* $S_k$ for $S_i$ if

$$\Pr[S_j = s \mid S_i = s] - \Pr[S_k = s \mid S_i = s] > 0 \text{ and}$$
$$\Pr[S_j = s' \mid S_i = s] - \Pr[S_k = s' \mid S_i = s] < 0$$

for all $s, s' \in \Omega$ with $s \neq s'$.

However, the guarantee for truth-telling (*informed truthfulness*) is weaker than the binary setting.

**Theorem A.2.** *Given any discrete domain $\Omega$, if for each agent $i$ the associated agent $j$'s signal uniformly dominates $k$'s signal for $i$'s signal (definition A.1), Mechanism 3's scheme is symmetrically informed truthful so that*

1. *truth-telling is a strict equilibrium, and*

2. *each agent's expected payment in truth-telling is no less than the payment in any other symmetric equilibria and strictly better than any uninformed equilibrium's.*

*Proof.* First truth-telling is a strict equilibrium, because if $S_i = s$,

$$\arg\max_{\hat{s}} \mathbb{E}\left[U^{BPP}(\hat{s}, S_j, S_k) \mid S_i = s\right]$$
$$= \arg\max_{\hat{s}} \Pr[S_j = \hat{s} \mid S_i = s] - \Pr[S_k = \hat{s} \mid S_i = s]$$
$$= s \qquad\qquad\qquad\qquad\qquad\qquad\qquad\qquad \text{(by definition A.1)}$$

Additionally, because $\Pr[S_j = s \mid S_i = s] - \Pr[S_k = s \mid S_i = s] > \Pr[S_j = s' \mid S_i = s] - \Pr[S_k = s' \mid S_i = s]$ for all $s' \neq s$, summing over all possible $s' \in \Omega$ on both sides gets $\Pr[S_j = s \mid S_i = s] - \Pr[S_k = s \mid S_i = s] > 0$ and

$$\mathbb{E}\left[U^{BPP}(S_i, S_j, S_k)\right] > 0.$$

For any informed equilibrium, by a direct computation $\mathbb{E}\left[U^{BPP}(\hat{S}_i, \hat{S}_j, \hat{S}_k)\right] = 0$.

Finally, we show that the truth-telling has the maximum expected payment for each agents. When all agent use a strategy $\sigma : \Omega \to \Omega$, agent $i$'s expected payment is

$$\sum_{s_i, \hat{s}_i \in \Omega} \Pr[S_i = s_i] \sigma(s_i, \hat{s}_i) \mathbb{E}\left[U^{BPP}(\hat{s}_i, \hat{S}_j, \hat{S}_k) \mid S_i = s_i\right]$$

$$= 2 \sum_{s_i, \hat{s}_i \in \Omega} \Pr[S_i = s_i] \sigma(s_i, \hat{s}_i) \sum_{s \in \Omega} (\Pr[S_j = s \mid S_i = s_i] - \Pr[S_k = s \mid S_i = s_i]) \sigma(s, \hat{s}_i)$$

$$= 2 \sum_{s_i \in \Omega} \Pr[S_i = s_i] \sum_{\hat{s}_i, s \in \Omega} \sigma(s_i, \hat{s}_i) \sigma(s, \hat{s}_i) (\Pr[S_j = s \mid S_i = s_i] - \Pr[S_k = s \mid S_i = s_i])$$

Let $f_{s_i}(s) := \sum_{\hat{s}_i \in \Omega} \sigma(s_i, \hat{s}_i) \sigma(s, \hat{s}_i)$ which is between 0 and 1, because $f_{s_i}(s) \leq \sum_{\hat{s}_i \in \Omega} \sigma(s_i, \hat{s}_i) \sum_{\hat{s}_i \in \Omega} \sigma(s, \hat{s}_i) = 1$. Then the expectation becomes

$$\sum_{s_i, \hat{s}_i \in \Omega} \Pr[S_i = s_i] \sigma(s_i, \hat{s}_i) \mathbb{E}\left[U^{BPP}(\hat{s}_i, \hat{S}_j, \hat{S}_k) \mid S_i = s_i\right]$$

$$= 2 \sum_{s_i \in \Omega} \Pr[S_i = s_i] \sum_{s \in \Omega} \left(\Pr[S_j = s \mid S_i = s_i] - \Pr[S_k = s \mid S_i = s_i]\right) f_{s_i}(s)$$

$$\leq 2 \sum_{s_i \in \Omega} \Pr[S_i = s_i] \left(\Pr[S_j = s_i \mid S_i = s_i] - \Pr[S_k = s_i \mid S_i = s_i]\right)$$

$$= \mathbb{E}\left[U^{BPP}(S_i, S_j, S_k)\right]$$

The inequality holds because $f_{s_i} \in [0, 1]$ and definition A.1. Therefore, we complete the proof. $\quad\square$

## B   Proofs in Section 2: Bayesian SST model and other models

The proofs of propositions 2.3 and 2.5 are standard, and variations can be found in related literature. We include proofs here for completeness.

*Proof of proposition 2.3.* First given $\theta \in \mathbb{R}^{\mathcal{A}}$, for all distinct $a, a', a'' \in \mathcal{A}$, $\Pr[T_\theta(a, a') = 1], \Pr[T_\theta(a', a'') = 1] > 1/2$ implies that $\theta_a - \theta_{a'} > 0$ and $\theta_{a'} - \theta_{a''} > 0$ becuase $F$ is strictly increasing and $F(0) = 1/2$. Because $\theta_a - \theta_{a''} = \theta_a - \theta_{a'} + \theta_{a'} - \theta_{a''} > \max(\theta_a - \theta_{a'}, \theta_{a'} - \theta_{a''})$, we have

$$\Pr[T_\theta(a, a'') = 1] = F(\theta_a - \theta_{a''})$$
$$> \max F(\theta_a - \theta_{a'}), F(\theta_{a'} - \theta_{a''})$$
$$= \max \Pr[T_\theta(a, a') = 1], \Pr[T_\theta(a', a'') = 1]$$

and thus $T_\theta$ is strongly stochastically transitive for all $\theta$ with distinct coordinates which happens surely as $\nu$ is non-atomic. Finally, since the distribution on $\theta$ is exchangeable on each coordinate, $\mathbb{E}\left[\mathbb{E}\left[T_\theta(a, a')\right]\right] = 0$ for all $a, a'$. $\quad\square$

*Proof of proposition 2.5.* First given $\theta \in \Theta$, for all distinct $a, a' \in \mathcal{A}$, if the rank of $a$ is higher than $a'$,

$$\Pr[T_\theta(a, a') = 1] = h_\eta(\theta(a') - \theta(a) + 1) - h_\eta(\theta(a') - \theta(a))$$

where $h_\eta(x) = \frac{x}{1 - \exp(-\eta x)}$ by Busa-Fekete et al. [7].

**Claim B.1.** *For any $\eta > 0$ and $x \in \mathbb{Z}_{>0}$, the difference $h_\eta(x + 1) - h_\eta(x)$ is increasing and larger than $1/2$ where $h_\eta(x) = \frac{x}{1 - \exp(-\eta x)}$.*

By claim B.1, $\Pr[T_\theta(a, a') = 1], \Pr[T_\theta(a', a'') = 1] > 1/2$ implies that $\theta(a') - \theta(a) > 0$ and $\theta(a'') - \theta(a') > 0$. Thus, $\theta(a'') - \theta(a) > \max(\theta(a'') - \theta(a'), \theta(a'') - \theta(a'))$, and

$$\Pr[T_\theta(a, a'') = 1] = h(\theta(a'') - \theta(a) + 1) - h(\theta(a'') - \theta(a))$$
$$> \max h(\theta(a'') - \theta(a') + 1) - h(\theta(a'') - \theta(a')), h(\theta(a') - \theta(a) + 1) - h(\theta(a') - \theta(a))$$
$$= \max \Pr[T_\theta(a, a') = 1], \Pr[T_\theta(a', a'') = 1]$$

where the second inequality is due to claim B.1. Therefore, $T_\theta$ is strongly stochastically transitive for all $\theta$. Finally, $\mathbb{E}[\mathbb{E}[T_\theta(a, a')]] = 0$ for all $a, a'$ since $\theta$ is an uniform distribution on rankings. $\square$

*Proof of claim B.1.* We first prove that the function $h_\eta(x) = \frac{x}{1 - \exp(-\eta x)}$ is increasing and strictly convex on $x \geq 0$. Because $h_\eta(x) = \frac{1}{\eta} h_1(\eta x)$, for all $\eta, x$, it is sufficient to consider $\eta = 1$. First, $h_1'(x) = \frac{1 - (x+1)e^{-x}}{(1 - e^{-x})^2} > 0$, so $h_1$ is increasing. Second, as $h_1''(x) = \frac{e^{-x}((x-2) + (x+2)e^{-x})}{(1 - e^{-x})^3}$, to show $h_1''(x) > 0$ for all $x > 0$, it is sufficient to show that $g(x) = (x - 2) + (x + 2)e^{-x} > 0$. Because $g(0) = 0$ and $g'(x) = 1 - (x+1)e^{-x} > 0$, $g(x) > 0$ for all $x > 0$. Therefore, $h_1$ is strictly convex.

On the other hand, $h_\eta(x+2) - h_\eta(x+1) > h_\eta(x+1) - h_\eta(x)$ for all $x$ by convexity, and $h_\eta(2) - h_\eta(1) = \frac{1}{1 + e^{-\eta}} > \frac{1}{2}$ which completes the proof. $\square$

# C   Proofs in Section 3 and 4

## C.1   Uniform dominance from Bayesian SST

*Proof of lemma 4.2.* With a prior similar assumption for Bayesian SST model, we only need to show

$$\Pr[S(a'', a') = 1 \mid S(a, a') = 1] > \Pr[S(a'', a) = 1 \mid S(a, a') = 1], \tag{5}$$

and the other case $\Pr[S(a'', a') = -1 \mid S(a, a') = -1] > \Pr[S(a'', a) = -1 \mid S(a, a') = -1]$ follows by symmetry. To prove eq. (5), we can rewrite the conditional probability in expectations of $T_\theta$.

$$\Pr[S(a'', a') = 1 \mid S(a, a') = 1]$$
$$= \frac{\int \Pr[T_\theta(a'', a') = 1, T_\theta(a, a') = 1 \mid \theta]dP_\Theta}{\int \Pr[T_\theta(a, a') = 1 \mid \theta]dP_\Theta}$$
$$= \frac{\int \Pr[T_\theta(a'', a') = 1 \mid \theta]\Pr[T_\theta(a, a') = 1 \mid \theta]dP_\Theta}{\int \Pr[T_\theta(a, a') = 1 \mid \theta]dP_\Theta} \qquad \text{(conditional independent)}$$
$$= 2 \int \Pr[T_\theta(a'', a') = 1 \mid \theta]\Pr[T_\theta(a, a') = 1 \mid \theta]dP_\Theta \qquad \text{(a prior similar)}$$
$$= 2 \int \frac{\mathbb{E}[T_\theta(a'', a') \mid \theta] + 1}{2} \frac{\mathbb{E}[T_\theta(a, a') \mid \theta] + 1}{2} dP_\Theta \qquad \text{(binary value)}$$
$$= \frac{1}{2} \int \mathbb{E}[T_\theta(a'', a') \mid \theta]\mathbb{E}[T_\theta(a, a') \mid \theta] + \mathbb{E}[T_\theta(a'', a') \mid \theta] + \mathbb{E}[T_\theta(a, a') \mid \theta] + 1 dP_\Theta$$
$$= \frac{1}{2} \int \mathbb{E}[T_\theta(a'', a') \mid \theta]\mathbb{E}[T_\theta(a, a') \mid \theta] + 1 dP_\Theta. \qquad \text{(a prior similar)}$$

**Claim C.1.** *For any strongly stochastically transitive $T_\theta$ on $\mathcal{A}$, and distinct $a, a', a'' \in \mathcal{A}$*

$$\mathbb{E}[T_\theta(a, a') \mid \theta]\mathbb{E}[T_\theta(a'', a') \mid \theta] > \mathbb{E}[T_\theta(a, a') \mid \theta]\mathbb{E}[T_\theta(a'', a) \mid \theta].$$

With claim C.1, we have

$$\Pr[S(a'', a') = 1 \mid S(a, a') = 1] = \frac{1}{2} \int \mathbb{E}[T_\theta(a'', a') \mid \theta]\mathbb{E}[T_\theta(a, a') \mid \theta] + 1 dP_\Theta$$

$$> \frac{1}{2} \int \mathbb{E}[T_\theta(a'', a) \mid \theta]\mathbb{E}[T_\theta(a, a') \mid \theta] + 1 dP_\Theta = \Pr[S(a'', a) = 1 \mid S(a, a') = 1].$$

This completes the proof of eq. (5), and thus the uniform dominance. $\square$

*Proof of claim C.1.* We let $Q(\alpha, \alpha') := \mathbb{E}[T_\theta(\alpha, \alpha') \mid \theta] = 2\Pr[T_\theta(\alpha, \alpha') = 1 \mid \theta] - 1$ for all $\alpha, \alpha'$. Note that $Q(\alpha, \alpha') > 0$ if and only if $\Pr[T_\theta(\alpha, \alpha') = 1 \mid \theta] > 1/2$ and $Q(\alpha, \alpha') = -Q(\alpha', \alpha)$.

By symmetry, let $Q(a, a') > 0$. It is sufficient to show that

$$Q(a'', a') > Q(a'', a).$$

If $Q(a', a'') > 0$, by definition 2.1 $Q(a, a'') > Q(a', a'') > 0$ so $Q(a'', a') > Q(a'', a)$. Now consider $Q(a', a'') < 0$. If $Q(a'', a) < 0$, $Q(a'', a') > 0 > Q(a'', a)$. If $Q(a'', a) > 0$, we have $Q(a'', a) > 0, Q(a, a') > 0$, and thus $Q(a'', a') > Q(a'', a)$ by definition 2.1 □

## C.2 Uniform dominance and weak notions of stochastic transitivity

There are weaker forms of stochastic transitivity, raising the question of whether they are sufficient for uniform dominance as in lemma 4.2. We show that general weak stochastic transitivity is not sufficient. Additionally, we show that although the noisy sorting model from [5] is only weakly stochastically transitive but does not satisfy definition 2.1, it exhibits uniform dominance.

**Definition C.2** ([13]). A stochastic comparison function, $T : \mathcal{A}^2 \to \{-1, 1\}$, is *weakly stochastically transitive* if for all $a, a', a'' \in \mathcal{A}$ with $\Pr[T(a, a') = 1] > 1/2$ and $\Pr[T(a', a'') = 1] > 1/2$,

$$\Pr[T(a, a'') = 1] > 1/2.$$

Compared to definition 2.1, the weak stochastic transitivity only require the item $a$ is favorable than $a''$. Below we provide a simple weakly stochastically transitive example with a prior similar property that does not satisfy the uniform dominance in eq. (5).

**Example C.3.** Consider the set of three items and $\Theta$ consists of all ranking on $\mathcal{A}$ with uniform prior where $\theta$ maps each items to its value. Given $\theta \in \Theta$ so that if $\theta(a) > \theta(a') > \theta(a'')$,

$$\Pr[T_\theta(a, a') = 1] = \Pr[T_\theta(a', a'') = 1] = 0.9 \text{ and } \Pr[T_\theta(a, a'') = 1] = 0.6.$$

Note that the model is weakly stochastically transitive, because an item with a larger value is more favorable and the weak stochastic transitivity is reduced to transitivity on the values. However, the model is not strongly stochastically transitive, because $\Pr[T_\theta(a, a'') = 1] = 0.6 < \max\{\Pr[(T(a, a') = 1], \Pr[(T(a', a'') = 1]]\} = 0.9$. Finally, as the rank $\theta$ has a uniform prior, the model satisfies a prior similar assumption.

To conclude the example, we show that eq. (5) does not hold for the above model. By direct computation over all six possible ranking $\theta$, we have

$$\Pr[S(a'', a') = 1 \mid S(a, a') = 1]$$
$$= \frac{1}{2} \int \mathbb{E}[T_\theta(a'', a') \mid \theta] \mathbb{E}[T_\theta(a, a') \mid \theta] + 1 dP_\Theta$$
$$= \frac{1}{2}\left(1 - \frac{64}{6}\right),$$

but $\Pr[S(a'', a) = 1 \mid S(a, a') = 1] = \frac{1}{2}\left(1 + \frac{64}{6}\right)$. Therefore, we have $\Pr[S(a'', a') = 1 \mid S(a, a') = 1] < \Pr[S(a'', a) = 1 \mid S(a, a') = 1]$, and show that eq. (5) does not hold.

Though the above example shows that weak stochastic transitivity is not sufficient.[5] Below we show a popular weakly stochastically transitive model in Braverman and Mossel [5] has uniform dominance as in lemma 4.2.

**Example C.4.** Let $\Theta$ be the set of rankings on $\mathcal{A}$ and $\eta > 0$ be a parameter. Given a uniformly distributed reference ranking $\theta \in \Theta$, the noise ranking model [5] ensures that for all $\theta(a) > \theta(a')$

$$\Pr[T_\theta(a, a') = 1] = \frac{1}{2} + \eta$$

Note that the above model does not satisfy the strict inequality in definition 2.1, but by direct computation, $\Pr[S(a'', a') = 1 \mid S(a, a') = 1] = \frac{1}{2}\left(1 + \frac{4\gamma^2}{3}\right)$ and $\Pr[S(a'', a) = 1 \mid S(a, a') = 1] = \frac{1}{2}\left(1 - \frac{4\gamma^2}{3}\right)$, which satisfies lemma 4.2.

---

[5]In the above example, we can also decrease 0.9 to a smaller number that satisfies both uniform dominance and weak stochastic transitivity.

## C.3 Symmetrically strongly truthful from uniform dominance

*Proof of lemma 4.3.* Suppose $S_i = 1$. Because $\Pr[S_j = 1|S_i = 1] > \Pr[S_k = 1|S_i = 1]$, $\Pr[S_j = -1|S_i = 1] < \Pr[S_k = -1|S_i = 1]$. Therefore, $\arg\max_{\hat{s}\in\{-1,1\}} \Pr[S_j = \hat{s}|S_i = 1] - \Pr[S_k = \hat{s}_i|S_i = 1] = 1$. Identical argument holds for the case of $S_i = -1$ which completes the proof.

Additionally, the expected payment of truth-telling is

$$\mathbb{E}\left[U^{BPP}(S_i, S_j, S_k)\right] = \sum_a \Pr[S_i = s_i] \sum_{s_j,s_k} \Pr[S_j = s_j, S_k = s_k \mid S_i = s_i]U^{BPP}(s_i, s_j, s_k)$$

$$= 2\sum_a \Pr[S_i = s_i] \sum_{s_j,s_k} \Pr[S_j = s_j, S_k = s_k \mid S_i = s_i](\mathbf{1}[s_i = s_k] - \mathbf{1}[s_i = s_k])$$

$$= 2\sum_a \Pr[S_i = s_i]\left(\Pr[S_j = s_i \mid S_i = s_i] - \Pr[S_k = s_i \mid S_i = s_i]\right)$$

$$> 0$$

The last inequality holds due to definition 4.1. □

*Proof of lemma 4.4.* As $\sigma$ is uninformed, let $\mu(s) = \sigma(s,s) = \sigma(-s,s)$ and $\mu(-s) = \sigma(s,-s) = \sigma(-s,-s)$ for all $s$.

$$\mathbb{E}\left[U^{BPP}(\hat{s}_i, \hat{S}_j, \hat{S}_k) \mid S_i = s_i\right] = \sum_{\hat{s}_j,\hat{s}_k} \mu(\hat{s}_j)\mu(\hat{s}_k)U^{BPP}(\hat{s}_i, \hat{s}_j, \hat{s}_k) = \sum_{\hat{s}_j,\hat{s}_k} \mu(\hat{s}_j)\mu(\hat{s}_k)(\hat{s}_i\hat{s}_j - \hat{s}_i\hat{s}_k) = 0$$

The first equality holds as the reports are independent of signals. □

*Proof of lemma 4.5.*

$$\mathbb{E}_{P,\sigma}\left[U^{BPP}(\hat{s}_i, \hat{S}_j, \hat{S}_k) \mid S_i = s_i\right]$$

$$= \sum_{s_j,s_k,\hat{s}_j,\hat{s}_k} \Pr[S_j = s_j, S_k = s_k \mid S_i = s_i]\sigma(s_j, \hat{s}_j)\sigma(s_k, \hat{s}_k)U^{BPP}(\hat{s}_i, \hat{s}_j, \hat{s}_k)$$

$$= 2\sum_{s_j,s_k,\hat{s}_j,\hat{s}_k} \Pr[S_j = s_j, S_k = s_k \mid S_i = s_i]\sigma(s_j, \hat{s}_j)\sigma(s_k, \hat{s}_k)\left(\mathbf{1}[\hat{s}_i = \hat{s}_j] - \mathbf{1}[\hat{s}_i = \hat{s}_k]\right)$$

(by eq. (1))

$$= 2\sum_{s_j,\hat{s}_j} \Pr[S_j = s_j \mid S_i = s_i]\sigma(s_j, \hat{s}_j)\mathbf{1}[\hat{s}_i = \hat{s}_j] - 2\sum_{s_k,\hat{s}_k} \Pr[S_k = s_k \mid S_i = s_i]\sigma(s_k, \hat{s}_k)\mathbf{1}[\hat{s}_i = \hat{s}_k]$$

$$= 2\sum_{s,\hat{s}}(\Pr[S_j = s \mid S_i = s_i] - \Pr[S_k = s \mid S_i = s_i])\sigma(s, \hat{s})\mathbf{1}[\hat{s}_i = \hat{s}] \quad \text{(renaming dummy variables)}$$

$$= 2\sum_s(\Pr[S_j = s \mid S_i = s_i] - \Pr[S_k = s \mid S_i = s_i])\sigma(s, \hat{s}_i)$$

Let $\delta = \Pr[S_j = s_i \mid S_i = s_i] - \Pr[S_k = s_i \mid S_i = s_i] > 0$, because $S_j$ uniformly dominates $S_k$ for $S_i$. Additionally, $\Pr[S_j = -s_i \mid S_i = s_i] - \Pr[S_k = -s_i \mid S_i = s_i] = 1 - \Pr[S_j = s_i \mid S_i = s_i] - 1 + \Pr[S_k = s_i \mid S_i = s_i] = -\delta$. We have

$$\mathbb{E}_{P,\sigma}\left[U^{BPP}(\hat{s}_i, \hat{S}_j, \hat{S}_k) \mid S_i = s_i\right]$$

$$= 2\sum_s(\Pr[S_j = s \mid S_i = s_i] - \Pr[S_k = s \mid S_i = s_i])\sigma(s, \hat{s}_i)$$

$$= 2\delta\left(\sigma(s_i, \hat{s}_i) - \sigma(-a, \hat{s}_i)\right),$$

so $\arg\max_{\hat{s}_i\in\{-1,1\}} \mathbb{E}_{P,\sigma}\left[U^{BPP}(\hat{s}_i, \hat{S}_j, \hat{S}_k) \mid S_i = s_i\right] = \arg\max_{\hat{s}_i\in\{-1,1\}} \{\sigma(s_i, \hat{s}_i) - \sigma(-s_i, \hat{s}_i)\}$ which completes the proof. □

# D  Proofs in Section 5.1

Before diving into the proof, we introduce some notations. We further introduce Ising models with bias parameter $\alpha \in \mathbb{R}_{\geq 0}^V$ in addition to $\beta$ where

$$H(\mathbf{s}) = \sum_{i,j\in V} \beta_{i,j}s_is_j + \sum_{i\in V} \alpha_i s_i$$

and $\Pr_{\alpha,\beta}[\mathbf{S} = \mathbf{s}] \propto \exp(H(\mathbf{s}))$, for all configuration $\mathbf{s}$. Given $i \in V$, let the expectation and ratio be

$$\nu_i(\alpha, \beta) = \mathbb{E}_{\alpha,\beta}[S_i] = \Pr_{\alpha,\beta}[S_i = 1] - \Pr_{\alpha,\beta}[S_i = -1] \text{ and } \rho_i(\alpha, \beta) = \frac{\Pr_{\alpha,\beta}[S_i = 1]}{\Pr_{\alpha,\beta}[S_i = -1]}$$

respectively which are monotone to each other. We will omit $\alpha, \beta$ when clear. Given a subset $U \subseteq V$, $\mathbf{s}_U \in \{-1, 1\}^U$ is a configuration over the nodes in $U$, and $\mathbf{s}_U = 1$ if $x_\iota = 1$ for all $\iota \in U$. We write $\Pr[\cdot | \mathbf{S}_U = \mathbf{s}_U]$, $\nu_{i|\mathbf{S}_U = \mathbf{s}_U}$, and $\rho_{i|\mathbf{S}_U = \mathbf{s}_U}$ for the conditional probability, expectation and ratio when the configuration in $U$ is fixed as specified by $\mathbf{s}_U$.

**A lower bound for LHS**  Informally, we want to lower bound the correlation between adjacent $i$ and $j$ (friends). Note that as we remove edges (setting coordinates of $\beta$ to zeros), the correlation should decrease, and the smallest correlation between neighboring nodes $i$ and $j$ happens when $E = \{(i, j)\}$. Lemma D.2 formalizes this idea using the following monotone inequality [44, Theorem 17.2].

**Theorem D.1** (Griffiths' inequality). *For any $i \in V$, $\nu_i(\alpha, \beta) = \mathbb{E}_{\alpha,\beta}[S_i]$ is non-negative and non-decreasing in all $\beta_{j,k} \geq 0$ and $\alpha_j \geq 0$ with $j, k \in V$.*

**Lemma D.2.** *Given $V$ and $i, j \in V$, for all $\alpha, \beta$, and $\beta'$, if $\beta'_e = \beta_e$ when $e = (i, j)$ and $\beta'_e = 0$ otherwise, we have*

$$\nu_{i|S_j=1}(\alpha, \beta) \geq \nu_{i|S_j=1}(\alpha, \beta') \text{ and } \rho_{i|S_j=1}(\alpha, \beta) \geq \rho_{i|S_j=1}(\alpha, \beta').$$

*Proof.* First, note that we can write the conditional expectation $\mathbb{E}_{\alpha,\beta}[S_i \mid S_j = 1]$ as marginal expectation. Formally, consider $\alpha^\eta$ so that $\alpha^\eta_\iota = \alpha_\iota$ if $\iota \neq j$ and $\alpha^\eta_j = \alpha_j + \eta$. Because $\eta \to \alpha^\eta$ is non-decreasing, $\eta \to \nu_i(\alpha^\eta, \beta)$ is non-decreasing by theorem D.1. In addition, $\Pr_{\alpha^\eta,\beta}[S_i \mid S_j = s] = \Pr_{\alpha,\beta}[S_i \mid S_j = s]$ for all $s$, and $\Pr_{\alpha^\eta,\beta}[S_j = -1] = O(e^{-2\eta})$, so

$$\nu_{i|S_j=1}(\alpha, \beta) = \mathbb{E}_{\alpha,\beta}[S_i \mid S_j = 1] = \lim_{\eta \to +\infty} \nu_i(\alpha^\eta, \beta).$$

Similarly,

$$\nu_{i|S_j=1}(\alpha, \beta') = \mathbb{E}_{\alpha,\beta'}[S_i \mid S_j = 1] = \lim_{\eta \to +\infty} \nu_i(\alpha^\eta, \beta').$$

On the other hand, consider $\beta^\lambda$ so that $\beta^\lambda_e = \beta_e$ if $e \neq (i, j)$ and $\beta^\lambda_{i,j} = \beta_{i,j} + \lambda$. By theorem D.1, $\nu_i(\alpha^\eta, \beta^\lambda)$ is non-decreasing in $\lambda$ for all $\eta$. Because $\beta^0 = \beta'$ and $\beta^1 = \beta$, we have

$$\nu_{i|S_j=1}(\alpha, \beta') = \lim_{\eta \to +\infty} \nu_i(\alpha^\eta, \beta') \leq \lim_{\eta \to +\infty} \nu_i(\alpha^\eta, \beta) = \nu_{i|S_j=1}(\alpha, \beta)$$

Because $\rho_i = \frac{1+\nu_i}{1-\nu_i}$ is monotone in $\nu_i$, the second part follows. □

Given $\beta'$ defined in lemma D.2, by some direct computation with $\alpha = 0$

$$\rho_{i|S_j=1}(\alpha, \beta) \geq \rho_{i|S_j=1}(\alpha, \beta') = e^{2\alpha_i + 2\beta_{i,j}} = e^{2\underline{\beta}}. \tag{6}$$

**An upper bound for RHS**  Now, we need to upper bound the correlation between non-adjacent $i$ and $k$ (non-friends). We will use Weitz's self-avoiding walks reduction [65] to upper bound the correlation on general graph $G$ by the correlation on trees.

Given a general graph $G$, and an arbitrary node $i$, we can construct the Self Avoiding Walk Tree of $G$ rooted at $i$, denoted $T_{SAW}(G, i)$, so that $\Pr[S_i = 1 \mid \mathbf{S}_U = \mathbf{s}_U]$ is the same in $G$ as in the tree. We outline the construction. $T_{SAW}(G, i)$ enumerates all self-avoiding walks in $G$ starting at $i$ which terminates when it revisits a previous node (closes a cycle). Then, $T_{SAW}(G, i)$ introduces a leaf with a certain boundary condition. The self-avoiding walk never revisits a node immediately, so there all the leaves with fixed boundary conditions are at least three hops away from node $i$. Note that if $G$ has maximum degree $d$, $T_{SAW}$ is a $d$-ary tree.

**Theorem D.3** ([65]). *For any $\alpha, \beta$, node $i \in V$, and configuration $\mathbf{s}_U$ on $U \subset V$,*

$$\Pr_{\alpha,\beta}[S_i = 1 \mid \mathbf{S}_U = \mathbf{s}_U] = \Pr_{T_{SAW}(G,i)}[S_i = 1 \mid \mathbf{S}_U = \mathbf{s}_U].$$

First, with the above theorem, we have $v_{i|S_k=1}(\alpha, \beta) = \mathbb{E}_{\alpha,\beta}[S_i \mid S_k = 1] = \mathbb{E}_{T_{SAW}(G,i)}[S_i \mid S_k = 1]$. By the monotone property in theorem D.1, setting all two-hop neighbors $U$ in $T_{SAW}(G,i)$ to 1 (recalled that any boundary conditions for $T_{SAW}(G,i)$ being at least three hops away) increases the conditional expectation,

$$\mathbb{E}_{T_{SAW}(G,i)}[S_i \mid S_k = 1] \leq \mathbb{E}_{T_{SAW}(G,i)}[S_i \mid \mathbf{S}_U = 1, S_k = 1].$$

Let $T$ be the tree by truncating $T_{SAW}(G,i)$ at level 2. By the Markov property of Ising models, the expectation is equal to the expectation on $T$.

$$\mathbb{E}_{\alpha,\beta}[S_i \mid S_k = 1] \leq \mathbb{E}_{T_{SAW}(G,i)}[S_i \mid \mathbf{S}_U = 1] = \mathbb{E}_T[S_i \mid \mathbf{S}_U = 1]. \tag{7}$$

Finally, we can recursively compute the probability ratio $\rho_i$ (and thus expectation $v_i$) on trees. Specifically, given a rooted tree $T'$, we define $\rho_{T'}$ as the ratio of probabilities for the root to be $+1$ and $-1$ respectively, and $\rho_{T'|\mathbf{S}_U=\mathbf{s}_U}$ for the ratio of conditional probabilities. As stated in the following lemma, it is well known (see, for example, [22]) that the ratio of each node can be computed recursively over the children's ratio.

**Lemma D.4.** *Given a tree $T$ rooted at $i$ with parameter $(\alpha, \beta)$ and boundary condition $\mathbf{s}_U$,*

$$\rho_{T|\mathbf{S}_U=\mathbf{s}_U} = e^{2\alpha_i} \prod_{l=1}^{d} \frac{\rho_{T_l|\mathbf{S}_U=\mathbf{s}_U} e^{2\beta_{i,j_l}} + 1}{e^{2\beta_{i,j_l}} + \rho_{T_l|\mathbf{S}_U=\mathbf{s}_U}}$$

*where $j_1, \ldots, j_d$ are children of $i$ and $T_l$ is the subtree rooted at $j_l$ for all $l$.*

By the monotone property in theorem D.1, the maximum of right-hand side of eq. (7) happens when $T$ is a complete $d$-ary tree with $\beta = \overline{\beta}$. Therefore,

$$\rho_{i|S_k=1}(\alpha, \beta) \leq \left( \frac{e^{2(d+1)\overline{\beta}} + 1}{e^{2\overline{\beta}} + e^{2d\overline{\beta}}} \right)^d. \tag{8}$$

Finally, with eqs. (6) and (8), we have $\rho_{i|S_j=1}(\alpha, \beta) \geq e^{2\underline{\beta}} \geq \left( \frac{e^{2(d+1)\overline{\beta}}+1}{e^{2\overline{\beta}}+e^{2d\overline{\beta}}} \right)^d \geq \rho_{i|S_k=1}(\alpha, \beta)$ which implies eq. (4).

**Remark D.5.** Note that for any graph $G$ there exists small enough $\overline{\beta}, \underline{\beta}$ so that the condition in theorem 5.1 is satisfied, because the inequality become equality when $\overline{\beta} = \underline{\beta} = 0$, and $\frac{\partial}{\partial \beta} \frac{2\beta}{d} > 0 = \frac{\partial}{\partial \beta} \ln \frac{e^{2(d+1)\beta}+1}{e^{2\beta}+e^{2d\beta}}$.

The bound between $\beta$ and $d$ is necessary as shown in fig. 3. On the other hand, by the Markov property of the Ising model, the majority of all neighbor's signals is a sufficient statistic, and we can show the majority of all neighbor's signals are uniformly dominant to a non-neighbor's signal. Therefore, we can get a symmetrically strongly truthful mechanism by replacing $j$'s reports with the majority of reports from $i$'s neighbors.

# E  Proof of Theorem 5.3

The sufficient condition is done by lemma 4.3, because

$$\underset{\hat{s}_i \in \{-1,1\}}{\arg\max} \mathbb{E}\left[ \lambda U^{BPP}(\hat{s}_i, S_j, S_k) + \mu(S_j, S_k) \mid S_i = s_i \right]$$

$$= \underset{\hat{s}_i \in \{-1,1\}}{\arg\max} \mathbb{E}\left[ \lambda U^{BPP}(\hat{s}_i, S_j, S_k) \mid S_i = s_i \right]$$

$$= \underset{\hat{s}_i \in \{-1,1\}}{\arg\max} \mathbb{E}\left[ U^{BPP}(\hat{s}_i, S_j, S_k) \mid S_i = s_i \right] \qquad (\lambda > 0)$$

$$= s_i \qquad \text{(by lemma 4.3)}$$

For the necessary, given $U$, define $D(s_j, s_k) = \frac{1}{2}\left( U(1, s_j, s_k) - U(-1, s_j, s_k) \right)$ and $\mu(s_j, s_k) = \frac{1}{2}(U(1, s_j, s_k) + U(-1, s_j, s_k))$ for all $s_j$ and $s_k$ in $\{-1, 1\}$. Hence

$$U(s_i, s_j, s_k) = s_i \cdot D(s_j, s_k) + \mu(s_j, s_k), \forall s_i, s_j, s_k \in \{-1, 1\} \tag{9}$$

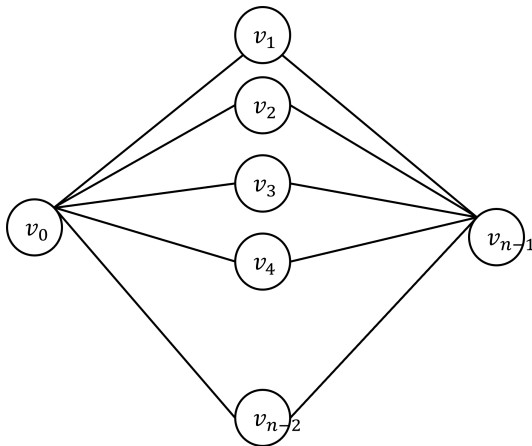

Figure 3: As fixing any $\beta, \bar{\beta}$, we can construct a simple graph with $V = \{v_0, \ldots, v_{n-1}\}$ and $E = \{(v_0, v_l), (v_l, v_{n-1}) : l = 1, \ldots, n-2\}$ where agent $v_0$ and $v_{n-1}$ are not connected but share $n-2$ common friends. We can show that the correlation between $S_0$ and $S_{n-1}$ converge to 1 as the number of common friends $d$ increases, while the correlation between $S_0$ and $S_1$ is bounded away from 1.

Given a joint distribution satisfying definition 4.1, we let $p^{s_i}(s_j, s_k) = \Pr[S_j = s_j, S_k = s_k \mid S_i = s_i]$ and additionally write $p^{s_i} = \begin{bmatrix} p^{s_i}(1,1) & p^{s_i}(1,-1) \\ p^{s_i}(-1,1) & p^{s_i}(-1,-1) \end{bmatrix}$. Then definition 4.1 ensures that

$$p^1(1,-1) > p^1(-1,1) \text{ and } p^{-1}(1,-1) < p^{-1}(-1,1).$$

Because $U$ is truthful for all uniformly dominant tuples, we have

$$
\begin{aligned}
0 &< \mathbb{E}\left[U(1, S_j, S_k) \mid S_i = 1\right] - \mathbb{E}\left[U(-1, S_j, S_k) \mid S_i = 1\right] = 2 \sum_{s_j, s_k} D(s_j, s_k) p^1(s_i, s_j) \\
0 &> \mathbb{E}\left[U(1, S_j, S_k) \mid S_i = -1\right] - \mathbb{E}\left[U(-1, S_j, S_k) \mid S_i = -1\right] = 2 \sum_{s_j, s_k} D(s_j, s_k) p^{-1}(s_i, s_j).
\end{aligned}
\tag{10}
$$

Suppose the following are true

$$D(1,-1) = -D(-1,1) > 0 \tag{11}$$
$$D(1,1) = D(-1,-1) = 0 \tag{12}$$

Let $\lambda = D(1,-1) > 0$. By eqs. (11) and (12), we have

$$
\begin{aligned}
U(s_i, s_j, s_k) &= s_i \cdot D(s_j, s_k) + \mu(s_j, s_k) & \text{(by eq. (9))} \\
&= \lambda \cdot s_i(s_j - s_k) + \mu(s_j, s_k) & \text{(by eqs. (9) and (11))}
\end{aligned}
$$

which completes the proof. Thus, we will construct three joint distributions satisfying definition 4.1 to prove eqs. (11) and (12).

The first joint distribution $p_1^{s_i}(s_j, s_k)$ with $0 < \delta \leq 1/2$

$$p^1 = \begin{bmatrix} 0 & 1/2 + \delta \\ 1/2 - \delta & 0 \end{bmatrix} \text{ and } p^{-1} = \begin{bmatrix} 0 & 1/2 - \delta \\ 1/2 + \delta & 0 \end{bmatrix}.$$

Then eq. (10) on the first distribution reduces to

$$0 < D(1,-1)p_1^1(1,-1) + D(-1,1)p_1^1(-1,1) = \frac{1}{2}(D(1,-1) + D(-1,1)) + \delta(D(1,-1) - D(-1,1))$$

$$0 > D(1,-1)p_1^{-1}(1,-1) + D(-1,1)p_1^{-1}(-1,1) = \frac{1}{2}(D(1,-1) + D(-1,1)) - \delta(D(1,-1) - D(-1,1)).$$

As we take $\delta$ to zero, we prove $D(1,-1) = -D(-1,1)$. Then plugging in with nonzero $\delta$, we have $D(1,-1) > 0$ and complete the proof of eq. (11).

The second joint distribution $p_2^{s_i}(s_j, s_k)$ with $0 \leq \epsilon \leq 1$ is

$$p^1 = \begin{bmatrix} 1-\epsilon & \frac{3}{4}\epsilon \\ \frac{\epsilon}{4} & 0 \end{bmatrix} \text{ and } p^{-1} = \begin{bmatrix} 1-\epsilon & \frac{\epsilon}{4} \\ \frac{3\epsilon}{4} & 0 \end{bmatrix}.$$

With eq. (11), eq. (10) reduces to

$$0 < (1-\epsilon)D(1,1) + \frac{\epsilon}{4}(D(1,-1) - D(-1,1))$$
$$0 > (1-\epsilon)D(1,1) - \frac{\epsilon}{4}(D(1,-1) - D(-1,1)).$$

By taking $\epsilon$ to zero, we prove $D(1,1) = 0$. We can prove $D(-1,-1) = 0$ using the similar trick and complete the proof of eq. (12).

## F  Additional empirical results

### F.1  Comparison data

Here we test if the dataset satisfy transitivity property. We denote the proportion of rankings such that item $a$ is higher than item $a'$ in the dataset by $p_{a>a'}$. If $p_{a>a'} > 1/2$, $p_{a'>a''} > 1/2$, and $p_{a>a''} > 1/2$, we say the triple of items $\{a, a', a''\}$ empirically satisfies transitivity. If $p_{a>a'} > 1/2$, $p_{a'>a''} > 1/2$, and $p_{a>a''} > \max\{p_{a>a'}, p_{a'>a''}\}$, we say the triple of items $\{a, a', a''\}$ empirically satisfies strong transitivity. We first test the transitivity of the SUSHI subdataset selected in section 6.1. We find that 100% of the item triples empirically satisfy transitivity, and 69.17% of the item triples empirically satisfy strong transitivity. This suggests that our transitivity assumption for the comparison data is mostly aligned.

Moreover, we conducted an experiment on the entire SUSHI dataset without any selection criteria and demonstrated the results in fig. 4. Observe that the ECDF of payments from original human users also dominates the payments under the uninformed strategy and the unilateral deviating strategy. This is consistent with our experimental results in section 6.1. However, there are two minor difference. First the separation of truth-telling from the other two is slightly less prominent than fig. 1 with the selection criteria. This may be due to a slightly lower degree of transitivity across agents with different backgrounds. In particular, we found the average value of $p_{a>a''} - \max\{p_{a>a'}, p_{a'>a''}\}$ is 0.0559 without the selection criteria which is less than 0.0604 with the selection criteria in fig. 1. Second, the fraction of agents receiving positive payments is slightly higher than in fig. 1 (0.785 and 0.763 respectively). This aligned with or empirical (strong) transitively which are 1 and 0.7667 compared to the above 1 and 0.69117. Furthermore, we also conducted experiments on other groups of users by changing the selection criteria. Those interested can refer to fig. 5, fig. 6 and table 1 for the results, which further verify the effectiveness of our mechanism.

| Selection criteria | Number of users | Average utility | Fraction of positive utility |
|---|---|---|---|
| All (No selection) | 5000 | 0.138 | 78.5% |
| Female, 30-49, Kanto/Shizuoka | 249 | 0.137 | 76.3% |
| Male, 30-49, Kanto/Shizuoka | 185 | 0.167 | 82.2% |
| Female, 5-29, Kanto/Shizuoka | 146 | 0.175 | 84.2% |
| Female, 50+, Kanto/Shizuoka | 26 | 0.13 | 80.8% |
| Female, 30-49, Tohoku | 30 | 0.174 | 83.3% |
| Female, 30-49, Hokuriku | 23 | 0.105 | 69.6% |

Table 1: Summary of truth-telling utility in appendix F.1.

### F.2  Networked data

Alongside fig. 2, Figure 7 and table 2 present empirical results for the top five popular artists in the dataset, excluding Lady Gaga, who are Britney Spears, Rihanna, The Beatles, and Katy Perry. All these settings show similar results. However, the Beatles' data is less conclusive as the payment distribution under the uninformed strategy profile is close to the truth-telling. This observation is

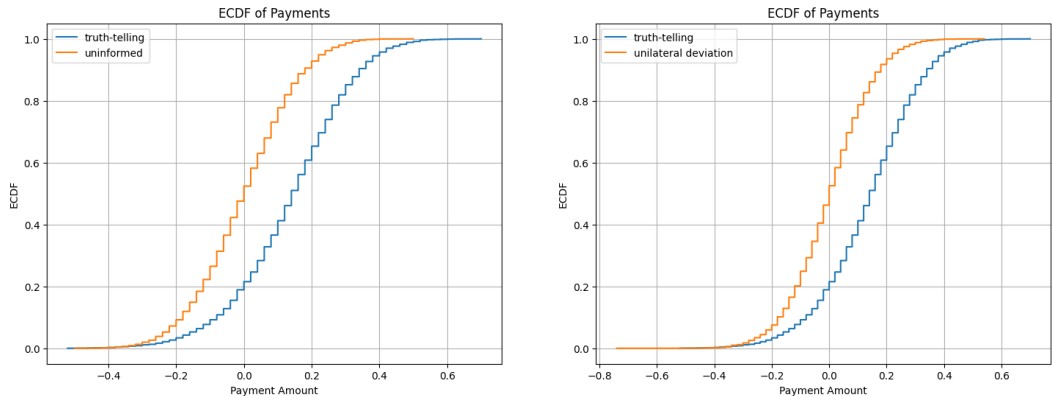

Figure 4: ECDF comparisons on all users without any selection.

also documented in Daskalakis et al. [12] which notes that the Ising model performs much better for rock artists than for pop artists. The authors conjecture that this may be due to the highly divisive popularity of pop artists like Lady Gaga and Britney Spears, whose listeners may form dense cliques within the graph.

Note that there is a buck of agent with a payment of around 0.5 under the truth-telling. This is because many non-listeners have no listener friends, and payment is $1 - [(1 - p) - p] = 2p$ is twice the popularity $p \approx 0.25$. Moreover, the jump is most minor for the Beatles, and indicates less agreement between non-listeners. Additionally, by the definition of bonus-penalty payment, we can see the payment of deviation is the minus of the truthful payment, so that the ECDF is symmetric around $(0, 0.5)$.

| Artists | Fraction of listener | Average utility | Fraction of positive utility |
|---|---|---|---|
| Lady Gaga | 32.2% | 0.37 | 76% |
| Britney Spears | 27.6% | 0.420 | 82.6% |
| Rihanna | 25.6% | 0.422 | 83.4% |
| The Beatles | 25.4% | 0.137 | 68.5% |
| Katy Perry | 25.0% | 0.361 | 79.9% |

Table 2: Summary of truth-telling utility in appendix F.2.

Figure 8 further shows the scatter plot of average payment and fraction of agents with positive payments across the top fifty popular artists where all settings have more than 60% percent of agents get positive payment. However, for less popular artists, the performance of our mechanism declines. This is expected, as we cannot provide effective incentives when only one agent listens to an artist.

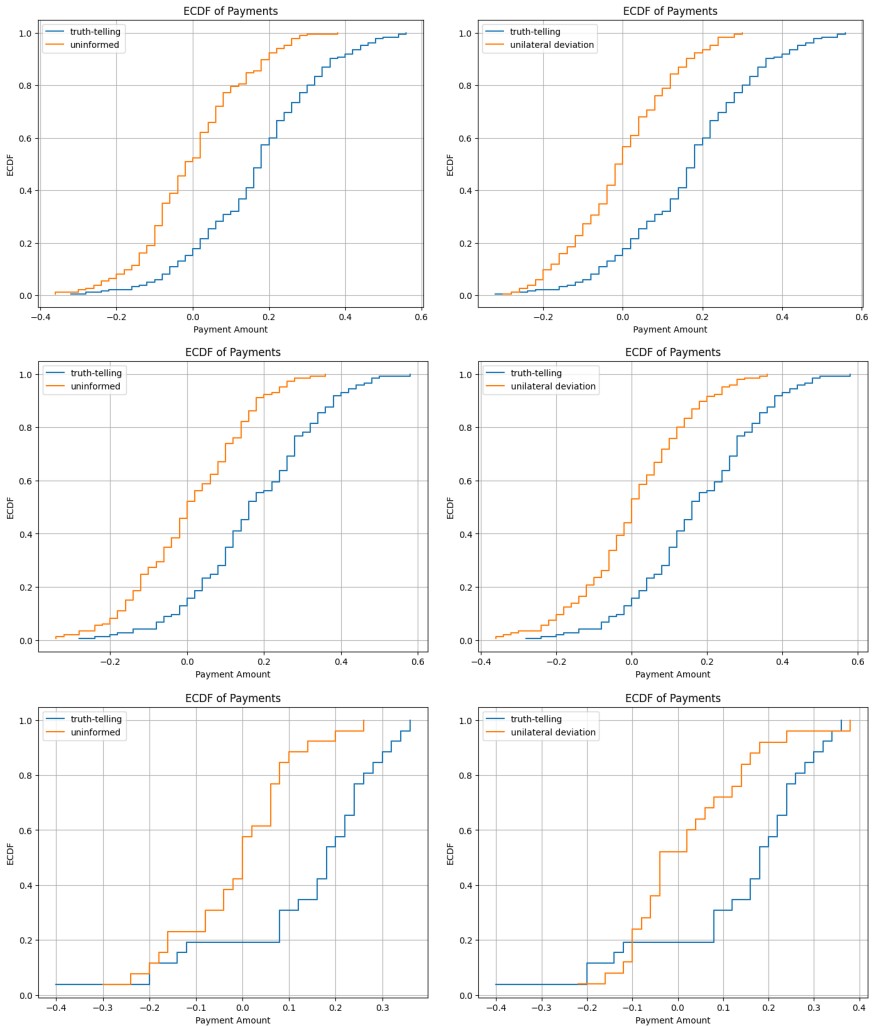

Figure 5: In each of the rows, we present the ECDF comparisons after changing the selection criteria for the user group as follows: from female to male, from ages 30–49 to ages 5–29, from ages 30–49 to ages 50+, respectively.

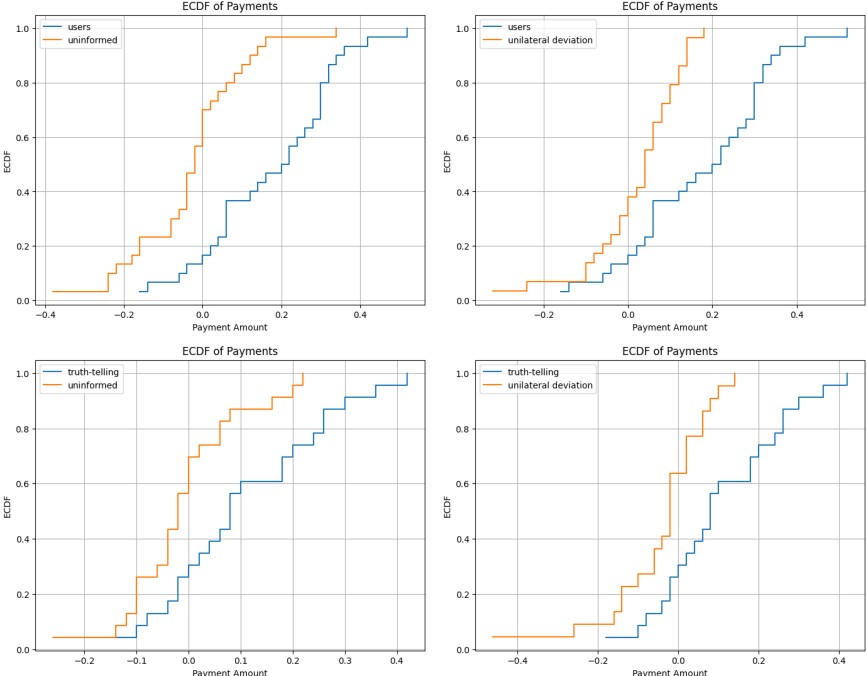

Figure 6: In each of the rows, we present the ECDF comparisons after changing the location criteria for the user group as follows: from mostly living in Kanto or Shizuoka to Tohoku until age 15, and from mostly living in Kanto or Shizuoka to Hokuriku until age 15, respectively.

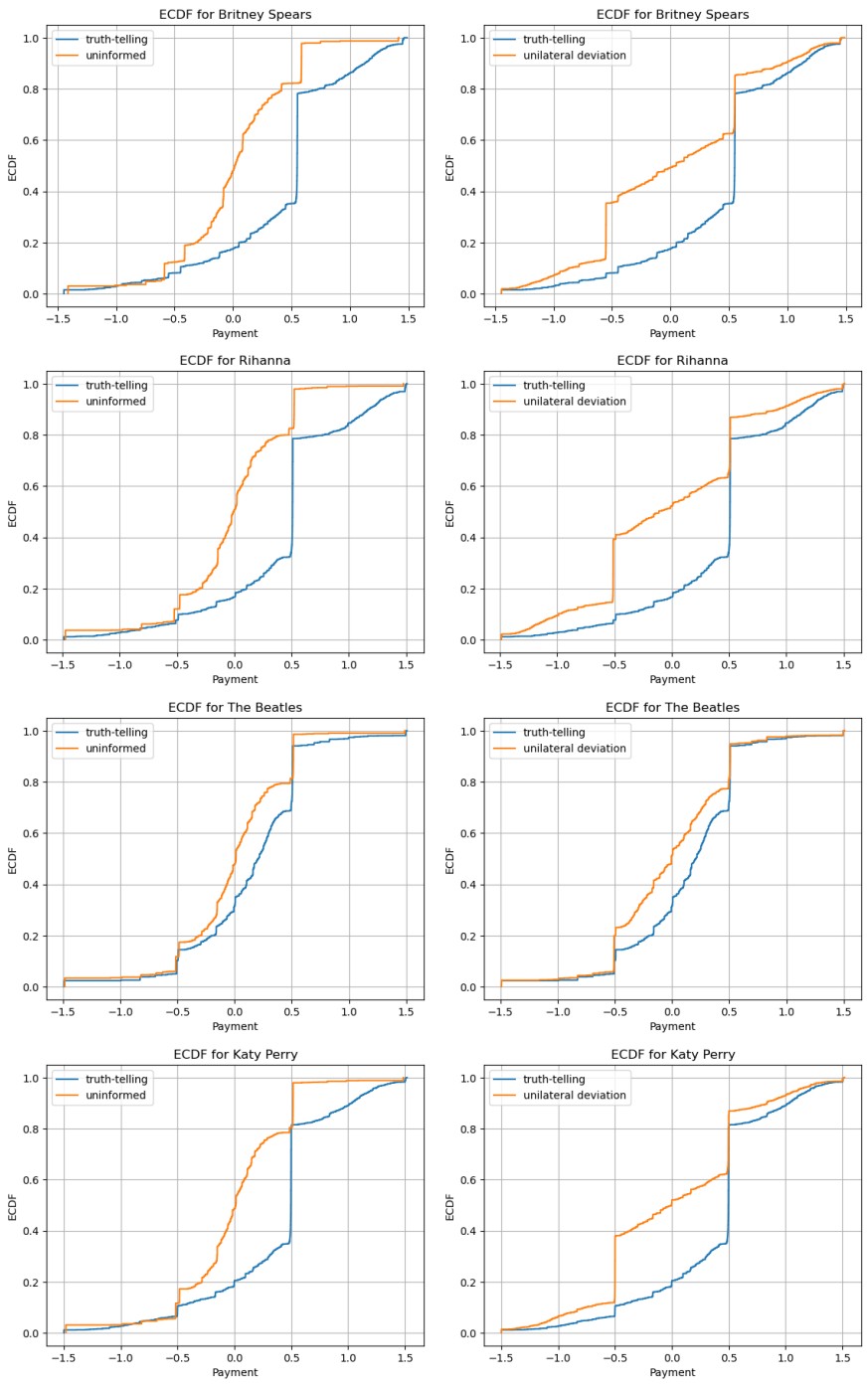

Figure 7: Last.fm dataset for other top five popular artists excluding Lady Gaga.

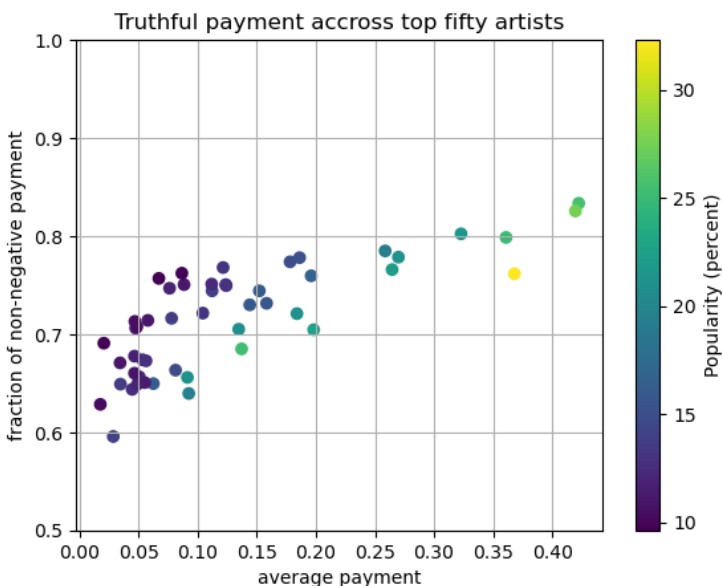

Figure 8: Average payment and fraction of positive payment under the truth-telling across top fifty popular artists.

