# OpenReview forum: "Carrot and Stick: Eliciting Comparison Data and Beyond"
_NeurIPS.cc/2024/Conference — NeurIPS 2024 poster_

### Official Review · Reviewer_A4wM · 2024-06-18

**Soundness:** 3
**Presentation:** 2
**Contribution:** 3
**Rating:** 6
**Confidence:** 2

**Summary:**

This paper studies elicting comparison data with truthfulness guarantees, specifically strict Bayesian Nash equilbrium. The authors utilize strong stochastic transitivity to define a Bayesian strong stochastic transitivity model and generalize several existing models. Under this model (i.e., problem setting), the authors propose a peer prediction mechanism based on bonus-penalty and show that it achieves symmetrically strong truthfulness, for both comparison data. Moreover, the authors also identify the conditions (in theoretical results) and the corresponding mechanism that achieves symmetric strong truthfulness for networked data and the general setting. Empirical results on two real-world datasets show that the proposed mechanism achieves better truthfulness or incentive, by showing that being truthful leads to a higher payment.

**Strengths:**

- The studied problem is important and relevant.
- The theoretical rigor is appreciated.
- The proposed approach seems sound, and the definition of Bayesian SST to generalize several existing models is interesting.

**Weaknesses:**

- An important requirement or assumption is the admissible condition to ensure that the assignment contains the necessary triplets. While the authors describe that one can create a superset, with a bounded size, to ensure the admissible condition. It appears that in doing so, one may need the agents to observe and report additional pairs of items (correct me if I am wrong). If so, in practice, one may not always be able to ask the agents to make addtional observations on required pairs of items (e.g., customers' reviews of products of services).

- The abstract and introduction motivates the importance of eliciting comparison data by citing several applications and use cases. However, the covered use cases by the empirical results seem to be less extensive than these.

**Questions:**

In lines 132-133,

> ... where the randomness of $S(\cdot, \cdot)$ comes from both $\theta$ and $T_{\theta}$
The randomness of $\theta$ is due to $\theta \sim P_{\Theta}$. What is the randomness of $T_{\theta}$? How should one interpret this, and its implications?

What other families of strategies are useful? Can strategically mis-behaving agents be accounted for, for instance to exploit this system to minimize someone's expected payment or intentionally aiming to obfuscate identification of the true comparison or ranking? In other words, are these considered families of strategies sufficient to consider some robustness to mis-behaving agents (and which types)?

In lines 173-174,

> ... she would expect that others will ... prefer $a$ over $a''$

What is the intuition behind this? Why would $a$ be preferred over $a''$ when the agent itself only has information that $a$ is preferred over $a'$?

In line 190-191,
> Hence, theorem 3.1 implies that agents’ manipulations can only decrease the probability of transitivity among their reports.

What is the implication of "decreasing the probability of transitivity among their reports"?

**Limitations:**

The method requires a specific problem setting or model called the Bayesian strong stochastic transitivity. But it is shown to generalize several other existing models.

---

> ### Author Rebuttal · Authors · 2024-08-07
>
> Thank you for your valuable input!
> ***
> **Question 1**: What is the randomness of $T_\theta$?
>
> **Answer**: $T_\theta$ is a stochastic comparison function that outputs random comparison given parameter $\theta$. The randomness of $T_\theta$ captures noise in observing comparisons. For instance, in the Bradley-Terry-Luce model, $\theta$ encodes each item's quality, and fixing the parameter $\theta$, the comparison outcome between two items $a$ and $a'$ is noisy and with level of randomness decided by $\theta_a-\theta_{a'}$.
> ***
> **Question 2**: External incentives: minimizes other's payment of obfuscate identification of the true ranking.
>
> **Answer**: Our setting focuses on agents who only care about their own payment and do not consider agents who have external incentives, for example, incentives to minimize someone else's expected payment or to obfuscate identification of the true comparison. Several previous works, however, offer potential solutions to handle such external incentives. For instance, as [49], we may incorporate robust statistics (or different privacy techniques) to mitigate the impact of a small group of agents attempting to attack one agent's payment or learning outcomes.
> ***
> **Question 3**: Are these considered families of strategies sufficient to consider some robustness to misbehaving agent?
>
> **Answer**: In our problem setting, since we work with binary reports, the only two pure
> strategies of the agent are truth-telling and flipping. Therefore, any mixed strategy of the agent can
> be represented by a linear combination of truth-telling and an uninformed strategy (and flipping).
> Consequently, our comparison of truth-telling and uninformed agents already indicates the higher
> payment of truthful agents over any other individual strategic behavior. Still, we add experiments of
> more complex group strategic behaviors to test the robustness of our mechanism. Please see the answer to question 6 in the rebuttal for Reviewer c3bN. Thank you!
>
> ***
> **Question 4**: Line 173-174. Why would $a$ be preferred over $a''$ when the agent itself only has information that $a$ is preferred over $a'$.}\yc{Can you check that this response is correct?
>
> **Answer**: When an agent observes that $a$ is preferred over $a'$, denoting this as $T(a,a')=1$, the agent considers two conditional probabilities for any given third item $a''$: $P[T(a'',a')=1|T(a,a')=1]$ and $P[T(a'',a)=1|T(a,a')=1]$. The first is the conditional probability that the third item $a''$ is preferred over the $a'$ and the second is the conditional probability that the third item $a''$ is preferred over $a$. Since the agent himself thinks $a$ is better than $a'$, it's more likely that the third item $a''$ is better than $a'$, than better than $a$. The first conditional probability is higher than the second conditional probability. We'll provide more explanation on this point in the paper.
> ***
> **Question 5**: Line 190-191. What is the implication of decreasing the probability of transitivity among their reports.
>
> **Answer**: Given three items $a,a',a''$ and three comparison outcomes $x = 1[a\prec a']$, $y = 1[a'\prec a'']$, and $z = 1[a''\prec a]$, these comparison outcomes satisfy transitivity if they form one of the $3!$ valid rankings. As agent's noisy comparisons are random, some comparison outcomes satisfy transitivity and some do not. The probability of transitivity is the probability that noisy comparison outcomes satisfy transitivity.
> ***
> **Question 6**: The method requires a specific problem setting or model called the Bayesian SST.
>
> **Answer**: We emphasize that Bayesian SST model is general. It is not a restriction for the comparison data application. Instead, it is a non-parametric generalization of several most commonly used parametric ranking models.
> ***
> **Question 7**: Admissible assignments.
>
> **Answer**: We do not require a single agent to compare additional pairs of items; instead, we only need that some other agents are assigned to those pairs.
> The admissible condition requires some control over the assignment $\mathcal{E}$.
> This is reasonable for the purpose of rank learning because these algorithms also need certain controls on $\mathcal{E}$. Otherwise, it would be impossible to rank an item if it is never compared to any other item. Moreover, several works on rank learning (e.g., [51] and Braverman and Mossel) require $\mathcal{E}$ to include all possible pairwise comparisons, which automatically satisfies our admissible condition.   Several platforms (including Amazon) actively encourage users to submit reviews on specific items, which can be considered a method to control $\mathcal{E}$.
> ***
> **Question 8**: Empirical results are less extensive.
>
> **Answer**: In the attached PDF, we run our mechanism on a new dataset: the HuggingFace H4 Stack Exchange Preference Dataset, which is a dataset used to align LLMs with human preferences. The experiment on this new dataset aligns with our introduction and abstract of improving the data quality of human preference training for LLMs. The dataset contains questions and their corresponding answers, each with voting data. In our experiment, we treat each vote as the report of an agent. For example, suppose a question has three answers, $a_1, a_2, a_3$, with vote numbers $v_1, v_2, v_3$. A vote (downvote) for $a_1$ means that the agent reports $a_2 \prec (\succ) a_1$ and $a_3 \prec (\succ) a_1$. Hence, there are $|v_1| + |v_2| + |v_3|$ agents in this example. We treat the original agents in the dataset as truth-telling agents. As in the experiment on the SUSHI dataset, we compare the payments of truth-telling agents with uninformed agents and unilateral strategies (see the first two figures). The ECDF of payments for truth-telling agents clearly dominates.
> ***
> **Reference**:
>
> Braverman, Mark, and Elchanan Mossel. "Noisy sorting without resampling." Proceedings of the
> nineteenth annual ACM-SIAM symposium on Discrete algorithms (SODA), 2008.

---

> > ### Comment · Reviewer_A4wM · 2024-08-09
> > **Post rebuttal**
> >
> > I thank the authors for their prepared response. Most of my questions are clarified and I am happy to maintain my current recommendation of weak accept.

---

> > > ### Author Response · Authors · 2024-08-10
> > >
> > > We thank the reviewer for acknowledging this!

---

### Official Review · Reviewer_oRu4 · 2024-07-12

**Soundness:** 4
**Presentation:** 4
**Contribution:** 4
**Rating:** 7
**Confidence:** 5

**Summary:**

This paper considers the problem of eliciting pairwise comparisons truthfully from strategic agents in the absence of ground truth. It considers the popular framework of peer prediction which is a class of mechanisms where reports from "peer" agents are used as a proxy for ground truth in order to design a payment scheme. The problem with using existing peer prediction mechanisms is that they mainly work for settings where tasks are drawn from the same distribution. However, this does not apply to the case of pairwise comparisons, as the two items being compared can be very different.

This paper designs a new peer prediction mechanism that is based on the Dasgupta-Ghosh mechanism where the agents also suffer a penalty in addition to rewards so as to avoid spurious "agree to agree" equilibria. The main novelty of the paper comes from the connection it makes to the well-known strong stochastic transitivity (SST) condition for pairwise comparison data. Under the assumption that pairwise comparisons satisfy SST condition, the mechanism cleverly utilizes this condition to ensure that the reward term exceeds the penalty term for truthful agents which is one of the main requirements for truthfulness. The paper also considers other elicitation tasks beyond pairwise comparisons, such as eliciting networked data, and give a general framework for extending.

**Strengths:**

In my opinion, the paper makes a good contribution to the literature on peer prediction. While the proposed mechanism is mainly based on the Dasgupta-Ghosh mechanism, it is a principled way of extending the Dasgupta-Ghosh mechanism to heterogenous tasks and utilizing task specific properties (such as SST) for incentive design. I really like that the paper connects the peer prediction literature to the ranking literature in a meaningful manner.

**Weaknesses:**

I am only concerned about the relevance of this paper to NeurIPS as this is mainly a mechanism design paper. But given the usefulness of pairwise comparisons in training LLMs, there is definitely need for truthful elicitation of comparison data.

The results on networked elicitation seem a bit disconnected from the results on elicitation of comparison data.

**Questions:**

Perhaps the framework also works for weak stochastic transitivity?

**Limitations:**

I did not find adequate discussion about limitations. Perhaps a section on limitations of the mechanism and peer prediction in-general might be useful.

---

> ### Author Rebuttal · Authors · 2024-08-07
>
> We greatly appreciate your valuable input!  Below, we will address your question in this paper.
> ***
> **Question**: Weak stochastic transitivity.
>
> **Answer**: This is an interesting question.  We conjecture that weak stochastic transitivity alone is not sufficient.  Our proof requires $\Pr[T_\theta(a'', a') = 1\mid T_\theta(a, a') = 1]$ is larger than $\Pr[T_\theta(a'', a) = 1\mid T_\theta(a, a') = 1]$, but weak stochastic transitivity only decides whether each term is greater than $1/2$ instead of magnitude.   For instance, we can make $\Pr[T_\theta(a'', a') = 1\mid \theta]$ always close to $1/2$ and $\Pr[T_\theta(a'', a) = 1\mid \theta]$ to almost one if $\Pr[T_\theta(a'', a) = 1\mid \theta]\ge 1/2$.  Thus, the conditional expectation $\Pr[T_\theta(a'', a) = 1\mid T_\theta(a, a') = 1]$ can be larger than $\Pr[T_\theta(a'', a') = 1\mid T_\theta(a, a') = 1]$, which will break our proof. That being said, our mechanism may still be symmetrically strongly truthful under Braverman and Elchanan's model, which is weakly stochastic transitive but not strongly stochastic transitive.
> ***
> **Reference**:
>
> Braverman, Mark, and Elchanan Mossel. "Noisy sorting without resampling." Proceedings of the nineteenth annual ACM-SIAM symposium on Discrete algorithms (SODA), 2008.

---

> > ### Comment · Reviewer_oRu4 · 2024-08-13
> >
> > Thank you for your response! I have no further questions and would like to keep my original score.

---

> > > ### Author Response · Authors · 2024-08-13
> > >
> > > Thank you for acknowledging this!

---

### Official Review · Reviewer_c3bN · 2024-07-12

**Soundness:** 3
**Presentation:** 2
**Contribution:** 2
**Rating:** 6
**Confidence:** 3

**Summary:**

* This paper proposes a peer prediction mechanism for elicitation of comparison data.
* In the model, there is a collection of items $A$, and a set of agents $N$ which privately observe noisy comparisons between items. Comparisons are characterized by a stochastic comparison function $T_\\theta$, parameterized by $\\theta \\in \\Theta$ and drawn from a common prior $P_\\Theta$.
* A peer prediction mechanism assigns a pair of items for each agent $i$ to compare. Each agent then observes a noisy comparison signal $s_i$ and strategically reports a possibly different value $\\hat{s}_i$ to maximize their ex-ante payment $\\mathbb{E} [M_i (\\hat{\\mathbf{s}})]$. The goal is to design a mechanism $M$ which elicits truthful reporting.
* The authors present Mechanism 1 for elicitation of comparison data, and prove that it is symmetrically strongly truthful (Theorem 3.1). The mechanism is based on a bonus-penalty payment function, which for agent $i$ awards agreement with some agent $j$, and penalizes agreement with some agent $k$.
* Section 5.1 presents a generalization of the mechanism to networked data, proving that it is symmetrically strongly truthful for signals sampled from the graph Ising model (Theorem 5.1). Section 5.2 further generalizes by showing a mechanism which is symmetrically strongly truthful for data with uniform dominance.
* Finally, empirical evaluation is performed on two datasets (Sushi preferences and Last.fm). Results seem to show stochastic dominance of payout in truthful reporting vs. random reporting.

**Strengths:**

* Presentation is clear and concise. Intuition is given for formal proofs.
* The concept of uniform dominance is interesting and seems to be generalizable.
* Theoretical results seem to be quite general, and applicable to many pairwise choice models.

**Weaknesses:**

* Empirical evaluation compares truthful reporting against benchmarks which seem relatively “simple” - uninformed random reporting, as opposed to more sophisticated forms of strategic behavior.
* Empirical evaluation procedure is not sufficiently clear. Method of constructing the graphs in the empirical section is not presented sufficiently formally. Graph legends and captions are uninformative.
* Implications of assumptions are not discussed in detail. Examples for such assumptions - assuming symmetric deviation, the ability to penalize users, assuming that items are a priori similar but ex-post distinct. See below.

**Questions:**

* What is the relation between Mechanism 3 and Mechanisms 1,2? Is it possible to think about Theorem 3.1 and Theorem 5.1 as corollaries of Theorem 5.2? If not, how do they fundamentally differ?
* “We assume items are a priori similar but ex-post distinct” L106 - What are the implications of this assumption? When does it apply, and does it restrict generality?
* For the given mechanism, is it possible to have a better BNE which is not symmetric? (e.g, a beneficial equilibrium where only some agents report truthfully?)
* What would be the implications of agents having limited liability? (e.g if the agents can choose not to participate in the game when they may get negative reward)
* Possible typo in L318: “ The figure shows that only about 50% of the agents using an uninformed random strategy receive positive payments, while over 75% of the users in the original dataset receive positive payments” - Percentages seem to add to more than 100%.

**Limitations:**

Results seem to rely on relatively strong assumptions, such as symmetric deviation, and the ability to penalize users. Despite that, such limitations don’t seem to be discussed in detail.

---

> ### Author Rebuttal · Authors · 2024-08-07
>
> Thank you for your valuable input.
> ***
> **Question 1**: What are the relations between Mechanism 3 and Mechanisms 1 and 2? Is it possible to think about Theorem 3.1 and Theorem 5.1 as corollaries of Theorem 5.2?
>
> **Answer**: Mechanism 3 offers a general scheme for designing peer prediction mechanisms, which requires finding uniformly dominant
> tuples. Identifying tuples that are uniformly dominant for different settings is a critical and nontrivial
> task. Mechanism 1 and mechanism 2 identify uniformly dominant tuples for the pairwise comparison
> setting under Bayesian SST model and networked data setting under the Ising model respectively.
> Theorem 3.1 and Theorem 5.1 can be viewed as instantiations of Theorem 5.2, with identifying and
> proving uniformly dominant tuples in the respective settings as a main contribution.
> ***
> **Question 2**: What are the implications of a priori similar but ex-post distinct?
>
> **Answer**: The a priori similar assumption implies that agents do not favor any item before observing their noisy comparison. Our
> mechanism allows agents to know the assignment $\mathcal{E}$ in advance. The a priori similar assumption
> means that, before observing the noisy comparison, agents do not have strict ordering of any pair of
> items even with the knowledge of the assignment $\mathcal{E}$. This assumption can be relaxed if we further
> randomize the assignment $\mathcal{E}$ by a uniform random permutation.
>
> The ex-post distinct assumption requires that the realized state $\theta$ is distinct for each item. Under
> the Bradley-Terry model (a special case of our Bayesian SST), this assumption means that no two
> candidates have the same realized scalar quality. The Mallow’s model (another special case of ours)
> always satisfies this assumption.
> ***
> **Question 3**: Limited liability? (e.g. if the agent can choose not to participate in the game when they may get
> negative reward)
>
> **Answer**: We do not require our mechanism to penalize users (i.e. give them
> a negative payment) because any positive affine transformation of the payment function preserves the
> equilibrium. For example, we can add a constant of 1 to the payment function in Equation (2). This
> ensures that the payment is either 2 or 0.
> ***
> **Question 4**: Possible typo: ...Percentages seem to add to more than 100%.
>
> **Answer**: Our experiment shows the
> histogram of agent’s payment under three different settings: (1) Truth-telling, where every agent uses
> the truthful strategy, we treat the original data as agents’ true preferences. (2) Uninformed, where
> every agent plays the same uninformed random strategy. (3) Unilateral deviation, where a single
> agent plays a non-truthful strategy while all other agents report truthfully.
>
> The numbers are for different settings and do not sum to 100%. We will rewrite the sentence to the
> following “The figure shows that in the uninformed random strategy setting only about 50% of the
> agents receive positive payments, while in the original dataset (truthful strategy setting) over 75% of
> the users receive positive payments”.
> ***
> **Question 5**: Symmetric deviation.
>
> **Answer**: Our symmetric strong truthfulness does not require symmetric deviation.
> The truth-telling strategy profile is a Bayesian Nash equilibrium where any individual’s unilateral
> deviation results in a worse payment for the agent. Additionally, we show that our mechanism
> ensures the truth-telling strategy profile is better (gives higher payoff) than any other symmetric
> strategy profile. Considering symmetric strategy profiles is reasonable because they do not require
> complicated coordination among all agents.
> ***
> **Question 6**: More sophisticated form of strategic behavior.
>
> **Answer**: In our problem setting, since we work with binary reports, the only two pure
> strategies of the agent are truth-telling and flipping. Therefore, any mixed strategy of the agent can
> be represented by a linear combination of truth-telling and an uninformed strategy (and flipping).
> Consequently, our comparison of truth-telling and uninformed agents already indicates the higher
> payment of truthful agents over any other individual strategic behavior. Still, we add experiments of
> more complex group strategic behaviors to test the robustness of our mechanism.
>
> In the attached PDF, we ran the experiment on all datasets with a new group strategic behavior to
> further stress test our mechanism. For each dataset, we randomly divided the truth-telling agents
> originally in the dataset into two groups of equal numbers. In the first group, the agents remained
> truthful, but in the second group, the agents were replaced by uninformed agents. We then mixed
> the two groups of agents and adopted our mechanism on the mixed group of agents. Finally, we
> compared the payments of agents in the two groups (see the last 3 figures for the results). The
> results show that the payments of truth-telling agents still dominate the payments of uninformed
> agents.
>
> Also, we run our experiments on a new dataset: HuggingFace H4 Stack Exchange Preference Dataset, a dataset used to align LLMs with human preferences. The dataset contains questions and their corresponding answers, each with voting data. In our experiment, we treat each vote as the report of an agent.  We compare the payments of truth-telling agents with the new group behavior (see the third figure). The ECDF of payments for truth-telling agents clearly dominates. Please find the detailed setting in our global rebuttal.
> ***
> **Question 7**: Is it possible to have a better BNE which is not symmetric?
>
> **Answer**: From the mechanism designer’s perspective, the truth-telling equilibrium is the best equilibrium for the purpose of information elicitation. We focus on equilibria with symmetric strategy profiles. Our results do not rule out the possibility that there is
> some asymmetric equilibrium leading to higher agent payoff. However, such asymmetric equilibrium
> may require complicated coordination and hence is difficult to reach.

---

> > ### Author Response · Authors · 2024-08-12
> >
> > We would greatly appreciate your response to our rebuttal, which, in our view, effectively addresses your main concerns. If there are lingering questions, we would gladly engage in a discussion.

---

> > > ### Comment · Reviewer_c3bN · 2024-08-14
> > >
> > > Thank you for the response! The rebuttal seems to address my main concerns. I increase my score to 6.

---

> > > > ### Author Response · Authors · 2024-08-14
> > > >
> > > > Thank you for checking our response!

---

### Author Rebuttal · Authors · 2024-08-07

Please find our additional experiment results attached. Thanks!

In the first three figures, we run our experiments, including the new group strategical behavior, on a new dataset: the HuggingFace H4 Stack Exchange Preference Dataset, which is a dataset used to align LLMs with human preferences. The dataset contains questions and their corresponding answers, each with voting data. For example, suppose a question has three answers, $a_1, a_2, a_3$, with vote numbers $v_1, v_2, v_3$. A vote (downvote) for $a_1$ means that the agent reports $a_2 \prec (\succ) a_1$ and $a_3 \prec (\succ) a_1$. Hence, there are $|v_1| + |v_2| + |v_3|$ agents in this example. We treat the original agents in the dataset as truth-telling agents. In our experiment, we treat each vote as the report of an agent.  As in the experiment on the SUSHI dataset, we compare the payments of truth-telling agents with uninformed agents, unilateral strategies, as well as the new group behavior (which is introduced in the rebuttal for Reviewer c3bN). The ECDF of payments for truth-telling agents clearly dominates. Due to limited computing resources, we only selected the first 100 questions in the dataset with at least three answers with nonzero votes.

The last three figures are the experiment of the new group behavior on our original datasets.
***
**Reference**:

Lambert, Nathan, Lewis Tunstall, Nazneen Rajani, and Tristan Thrush. "HuggingFace H4 Stack Exchange Preference Dataset." Hugging Face, 2023.

---

### Decision · Program_Chairs · 2024-09-25

**Decision:**

Accept (poster)

**Comment:**

This paper proposes a peer-prediction based mechanism for truthfully eliciting noisy pairwise comparisons from multiple agents whose preferences are drawn from a common prior, but not otherwise observable by the center.  The mechanism is proven truthful in a particularly strong sense (truthtelling is both strictly Bayes-Nash and the highest-payoff symmetric equilibrium).  Results are validated against a number of datasets from the literature.

The reviewers were uniformly positive about this paper.  Its results and new concepts are likely to be relevant beyond the specific setting of the paper.  There were some concerns about the need for strong assumptions, but these were largely addressed during the discussion period.